

# European NO$_x$ emissions in WRF-Chem derived from OMI: impacts on summertime surface ozone

Auke J. Visser[1], K. Folkert Boersma[1,2], Laurens N. Ganzeveld[1], and Maarten C. Krol[1,3]

[1]Wageningen University, Meteorology and Air Quality Section, Wageningen, the Netherlands
[2]Royal Netherlands Meteorological Institute, R&D Satellite Observations, de Bilt, the Netherlands
[3]Institute for Marine and Atmospheric Research Utrecht, Utrecht University, Utrecht, the Netherlands

**Correspondence:** Auke Visser (auke.visser@wur.nl)

**Abstract.** Ozone (O$_3$) is a secondary air pollutant that negatively affects human and ecosystem health. Ozone simulations with regional air quality models suffer from unexplained biases over Europe, and uncertainties in the emissions of ozone precursor group nitrogen oxides (NO$_x$ = NO + NO$_2$) contribute to these biases. The goal of this study is to use NO$_2$ column observations from the OMI satellite sensor to infer top-down NO$_x$ emissions in the regional meteorology-chemistry model WRF-Chem, and to evaluate the impact on simulated surface O$_3$ with in situ observations. We first perform a simulation for July 2015 over Europe and evaluate its performance against in situ observations from the AirBase network. The spatial distribution of mean ozone concencentrations is reproduced satisfactorily. However, the simulated maximum daily 8-hour ozone concentration (MDA8 O$_3$) is underestimated (mean bias error (MBE) = -14.2 μg m$^{-3}$), and its spread is too low. We subsequently derive satellite-constrained surface NO$_x$ emissions using a mass balance approach based on the relative difference between OMI and WRF-Chem NO$_2$ columns. The method accounts for feedbacks through OH, NO$_2$'s dominant daytime oxidant. Our optimized European NO$_x$ emissions amount to 0.50 Tg N (for July 2015) 0.18 Tg N higher than the bottom-up emissions (which lacked agricultural soil NO$_x$ emissions). Much of the increases occur across Europe, in regions where agricultural soil NO$_x$ emissions dominate. Our best estimate of soil NO$_x$ emissions in July 2015 is 0.1 Tg N, much higher than the bottom-up 0.02 Tg N natural soil NO$_x$ emissions from the MEGAN model. A simulation with satellite-updated NO$_x$ emissions reduces the systematic bias between WRF-Chem and OMI NO$_2$ (slope = 0.98, r$^2$ = 0.84), and reduces the low bias against independent surface NO$_2$ measurements by 1.1 μg m$^{-3}$ (-56%). Following these NO$_x$ emission changes, daytime ozone is strongly affected, since NO$_x$ emission changes particularly affect daytime ozone formation. Monthly averaged simulated daytime ozone increases by 6.0 μg m$^{-3}$, and increases of >10 μg m$^{-3}$ are seen in regions with large emission increases. With respect to the initial simulation, MDA8 O$_3$ has an improved spatial distribution, expressed by an increase in r$^2$ from 0.40 to 0.53, and a reduced mean bias (-7.4 μg m$^{-3}$, -48%). Overall, our results highlight the dependence of surface ozone on its precursor NO$_x$ and demonstrate that simulations of surface ozone benefit from constraining surface NO$_x$ emissions by satellite NO$_2$ column observations.

## 1 Introduction

Ozone (O$_3$) is an air pollutant that affects human and ecosystem health (Lelieveld et al., 2015; Ainsworth et al., 2012). It also affects radiative forcing directly as a greenhouse gas (IPCC, 2013), and indirectly by impacting ecosystem carbon uptake via





deposition (Sitch et al., 2007). Despite decreases in ozone concentrations in Europe since 2000 (Chang et al., 2017), peak ozone concentrations still exceed the WHO air quality guideline of 100 µg m$^{-3}$ and the European long-term objective of 120 µg m$^{-3}$ (EMEP/CCC, 2016). For example, 87% of European air quality stations did not meet this long-term objective (EEA, 2017) in 2015, and vegetation exposure thresholds were exceeded in large parts of the continent during this year, particularly
in Southern and Central Europe (Rouïl and Meleux, 2018).

The formation of ozone in the lower troposphere is a photochemical process that depends nonlinearly on concentrations of its precursor species nitrogen oxides ($NO_x = NO + NO_2$) and volatile organic compounds (VOCs) (e.g. Sillman et al., 1990). In $NO_x$-limited conditions, ozone production increases with $NO_x$ emissions and is less sensitive to VOC emissions. However, ozone production under $NO_x$-saturated conditions increases with VOC emissions, but decreases with increasing
$NO_x$ emissions. European $NO_x$ emissions are dominated by the anthropogenic contribution from fossil fuel combustion for transportation, electricity generation and industry. In summer, there are additional contributions from soils and lightning, which together comprise  40% of the total European $NO_x$ emission budget (Jaeglé et al., 2005). Soil $NO_x$ emissions in turn have an anthropogenic component, since nitrogen-containing fertilizers are partly re-emitted to the atmosphere as $NO_x$ (Steinkamp and Lawrence, 2011).

Anthropogenic emissions in Europe have decreased due to air pollution abatement measures and the economic crisis that started in 2008 (Castellanos and Boersma, 2012). Bottom-up anthropogenic emission inventories suggest a continued reduction of $NO_x$ emissions in more recent years. This is consistent with the ongoing development of European air quality conditions towards the $NO_x$-limited regime (Jin et al., 2017), which is projected to continue in the future (Beekmann and Vautard, 2010). On the other hand, a decrease in European anthropogenic and natural $NO_x$ emissions is not supported by trend analysis of
remote sensing and in situ $NO_2$ observations (Jiang et al., 2019, submitted), although this potentially reflects a growing relative contribution from natural $NO_x$ emission sources (Silvern et al., 2019). Nevertheless, downward anthropogenic emission trends have been suggested as an important driver of the decreasing trend in peak ozone concentrations in Europe (ETC/ACM, 2016).

Regional air quality (AQ) models are important tools for studying and forecasting ozone pollution. These models simulate processes relevant for ozone pollution at a resolution that can better capture observed spatial gradients compared to coarser
global models. Regional AQ models can therefore be applied to simulate polluted conditions in or surrounding urban areas, or for air quality impact assessments. Coupled (or "online") meteorology-chemistry models resolve meteorology, transport, chemical transformation and removal of pollutants at the same spatial and temporal resolution. The coupled treatment of meteorology and chemistry is mandatory, because ozone concentrations depend on feedbacks between meteorological and chemical processes: 1) $O_3$ sources such as chemical formation depend on radiation, temperature and water vapour (Pusede
et al., 2015; Coates et al., 2016), and 2) $O_3$ sinks, such as dry deposition, also largely depend on meteorological drivers (Clifton et al., 2017; Kavassalis and Murphy, 2017). However, coupled regional air quality models are subject to several sources of uncertainties. These uncertainties are related to the limited knowledge on ozone precursor emissoins (Kuenen et al., 2014; Pouliot et al., 2015), the representation of boundary conditions (Giordano et al., 2015), tropospheric chemistry in the chemical mechanism (Knote et al., 2015), and the land surface and its feedbacks with tropospheric chemistry (Baklanov et al.,
35  2014).



Many regional AQ models have been applied to simulate $NO_x$ and $O_3$ in European summers, for research and forecasting purposes. Models tend to underestimate summertime $NO_x$ compared to rural background in situ observations (Terrenoire et al., 2015; Mar et al., 2016). Comparison against satellite $NO_2$ column observations also revealed underestimations at regional scales (Huijnen et al., 2010; Aidaoui et al., 2015). Another study found both positive as well as negative biases, which were

attributed to the coarse resolution of the emission inventories (Pope et al., 2015). AQ models satisfactorily reproduce the spatial distribution in summer $O_3$. However, mean $O_3$ can be under- or overestimated depending on the model and chemical mechanism (Terrenoire et al., 2015; Mar et al., 2016). In addition, many models consistently underestimate peak ozone values that typically occur in the afternoon (Tuccella et al., 2012; Solazzo et al., 2012; Marécal et al., 2015; Im et al., 2015). This is problematic for air pollution impact assessments, since the peak ozone values are important for determining the detrimental

effects on human health and ecosystems.

The sensitivity of $O_3$ to its precursor $NO_x$, which is particularly pronounced in summer (e.g. Jin et al., 2017), suggests that there is good potential to improve $O_3$ simulations by constraining simulated $NO_x$ with observations. The past 20 years have seen the development of methods to estimate $NO_x$ emissions with satellite-based $NO_2$ columns in a mass balance approach, where biases in the model-simulated and satellite-observed $NO_2$ columns are used to update $NO_x$ emissions. The technique

has been applied in global models (Martin et al., 2003; Lamsal et al., 2008; Vinken et al., 2014a), and more recently also in regional models (e.g. Ghude et al., 2013). Applications of the technique include emission trend analysis (e.g. Lamsal et al., 2011) and source-specific constraints on $NO_x$ emissions (e.g. Ghude et al., 2013; Vinken et al., 2014a, b; Verstraeten et al., 2015). Changes in $NO_x$ emissions impact tropospheric chemistry, and therefore changes in $O_3$ are expected. This was shown by Ghude et al. (2013), who found local changes in surface $O_3$ mole fractions up to 10 ppb over India after satellite-based $NO_x$

emission scaling. Verstraeten et al. (2015) reported ozone increases up to 8 ppb at 800 hPa ($\pm 1.5$ km) in China after scaling local $NO_x$ emissions with OMI observations, and found that simulated free-tropospheric ozone between 3-9 km was in better agreement with tropospheric $O_3$ columns observed by the Tropospheric Emission Sounder. However, ozone changes at the surface after constraining $NO_x$ emissions with satellite observations have thus far not been evaluated with in situ data to our knowledge.

Considering the importance of $NO_x$ for simulations of ozone and the previously reported ozone changes after applying satellite-based $NO_x$ emissions, we here investigate the potential improvement in simulated surface ozone concentrations over Europe due to the application of satellite observations of $NO_2$ to adjust $NO_x$ emissions. To this end, we use the WRF-Chem meteorology-chemistry model (Grell et al., 2005) to simulate surface ozone in Europe in July 2015, at the approximate peak of the ozone season. We first perform a model evaluation with AirBase in situ $NO_2$ and $O_3$ observations (EEA, 2018) and OMI

$NO_2$ column measurements from the recently released QA4ECV dataset (Boersma et al., 2017a). We subsequently derive a new, OMI-based ("top-down") $NO_x$ emission inventory, and evaluate its effects on WRF-Chem simulations of surface $NO_2$ and $O_3$ with the independent AirBase observations.

The structure of the paper is as follows. We describe the model set-up and observations in section 2. Section 3 presents the method to calculate OMI-derived $NO_x$ emissions. In section 4, we evaluate a WRF-Chem set-up with bottom-up emissions

in situ and column observations, and in section 5 we describe the derived modified surface $NO_x$ emissions. We evaluate the





impacts on surface $NO_x$ and $O_3$ with independent in situ observations in section 6. We conclude with a discussion (section 7) and summarize our conclusions in section 8.

## 2 Model and data description

### 2.1 WRF-Chem

We perform simulations with the coupled meteorology-chemistry model WRF-Chem, version 3.7.1 (Grell et al., 2005). The model domain consists of 170 by 170 cells at $20 \times 20$ km$^2$ horizontal resolution covering Europe, centered at 51.98°N and 5.66°E. Vertically, the domain extends from the Earth's surface up to 50hPa, and consists of 27 layers with 13 layers in the lowermost 1500m. Chemistry simulations of $O_3$ and its precursor groups $NO_x$ and VOCs are performed with the CBM-Z gas-phase chemical mechanism (Zaveri and Peters, 1999). Simulations of atmospheric chemistry with this mechanism compare

well with the European multi-model mean for summer $O_3$ in a gas-phase mechanism comparison study (Knote et al., 2015). A complete list of parameterization options adopted in our WRF-Chem setup can be found in Table 1 of the Supplement. Our simulations were performed with a time stepping of 180 s for a period of 38 days (24 June - 31 July 2015), allowing a 1-week spin-up to analyze the model output for July. An evaluation of large-scale meteorological performance with ERA-Interim reanalysis fields can be found in Sect. 2 of the Supplement.

We used anthropogenic emissions from the TNO-MACC-III inventory (Kuenen et al., 2014) for 2011, the most recent inventory available when the model experiments were performed. TNO-MACC-III contains anthropogenic emissions for lumped species groups $NO_x$ and VOCs. $NO_x$ emissions were partitioned assuming that 97% is emitted as NO and 3% as $NO_2$. VOC emissions were divided over 15 emission categories in CBM-Z, following the VOC speciation by Archer-Nicholls et al. (2014). This speciation procedure is further described in Table 3 of the Supplement. Point source emissions were distributed over the

five lowermost model layers following sector-specific emission altitude profiles (Bieser et al., 2011).

Biogenic emissions of VOCs and soil $NO_x$ were calculated online with the MEGAN model implementation within WRF-Chem (Guenther et al., 2006, 2012). The domain-total biogenic isoprene emissions are 1.82 Tg of isoprene, which is slightly lower than the 9-year spread of 2-4.5 Tg isoprene for July, based on an inverse modeling study using OMI HCHO column measurements for 2005-2013 (Bauwens et al., 2016). We simulate lightning $NO_x$ emissions using a parameterization based on

cloud-top height (Price and Rind, 1993; Wong et al., 2013), using a flash rate of 80 mol flash$^{-1}$ based on a recent satellite-based estimate (Pickering et al., 2016). Simulations with higher flash rates of 500 mol flash$^{-1}$ (Ott et al., 2010) and 310 mol flash$^{-1}$ (Miyazaki et al., 2014) resulted in overestimated upper-tropospheric contributions to the $NO_2$ columns relative to OMI.

Anthropogenic emissions are the dominant $NO_x$ source over Europe in July with a total monthly emission strength of 304 Gg N (76%). Minor contributions are associated with lightning (81.4 Gg N; 20%) and soils (15.0 Gg N; 4%). We note that

especially soil $NO_x$ emissions are low compared to previous studies, in which soils, including agricultural areas, have been estimated to contribute $\pm 40$ % to the total European $NO_x$ emission budget (Jaeglé et al., 2005; Ganzeveld et al., 2010).

Meteorological initial and boundary conditions were taken from ERA-Interim reanalysis data (Dee et al., 2011). Chemical boundary conditions for $O_3$, NO, $NO_2$, CO and peroxyacetyl nitrate (PAN) are taken from the CAMS chemical reanal-



ysis product for Europe (Inness et al., 2015, retrieved at: http://apps.ecmwf.int/datasets/data/cams-nrealtime/levtype=sfc/).
Upper boundary conditions for ozone were prescribed with climatological values (retrieved at: https://www2.acom.ucar.edu/
wrf-chem/wrf-chem-tools-community).

## 2.2 AirBase $NO_2$ and $O_3$ in situ measurements

Surface measurements are taken from the European Air Quality Data Portal operated by the European Environment Agency,
hereafter referred to as AirBase (EEA, 2018). We used all data at rural background stations from the validated E1a data
stream. The large availability of the data allows us to make a strict selection on data availability. For monthly averages, we
discard stations if data is missing for more than 24 hours. Stations used for the evaluation of monthly averages at 12:00 h
UTC may have a maximum data gap of 1 data point. This resulted in a final selection of 184-397 stations, depending on the
performance metric (see Table 1). In our analysis of $O_3$ and $NO_2$ we evaluate monthly time series and mid-day (12:00 h UTC)
concentrations (denoted as $[O_3]^{12h}$ and $[NO_2]^{12h}$, respectively). We additionally calculate the maximum daily 8-hour mean
ozone concentration (MDA8 $O_3$), a widely applied metric for $O_3$ health impacts.

## 2.3 OMI $NO_2$ column measurements

We use tropospheric $NO_2$ columns from the Ozone Monitoring Instrument (OMI) onboard NASA's EOS Aura mission (Levelt
et al., 2006). The polar-orbiting instrument detects radiation backscattered from the Earth's atmosphere. Retrieval of tropo-
spheric vertical column densities (VCDs) from space follows a three-step procedure (Boersma et al., 2018). First, total slant
columns (SCDs; i.e., columns along the average light path through the atmosphere) are obtained from a spectral fit to the OMI-
measured reflectance spectra in the visible wavelength range using the Differential Optical Absorption Spectroscopy (DOAS)
method. Then, the stratospheric contribution component is separated from the total $NO_2$ column via data assimilation into the
TM5 global Chemistry Transport Model (Dirksen et al., 2011). The final step is to obtain tropospheric VCDs by dividing the
SCDs by a tropospheric Air Mass Factor (AMF) that describes the vertical sensitivity of the instrument to atmospheric $NO_2$
(Eskes and Boersma, 2003). This is a function of satellite viewing geometry, surface albedo, terrain height, cloud properties,
and a priori $NO_2$ profile.

The recent EU FP7 project Quality Assurance for Essential Climate Variables (QA4ECV) has led to the development of a
new OMI $NO_2$ data product (Boersma et al., 2017a). The underlying consortium retrieval algorithm is based on the $NO_2$ column
retrieval principles described in Boersma et al. (2007), but with improvements in the three aforementioned steps (Boersma et al.,
2018). Zara et al. (2018) described how better wavelength calibration, and inclusion of liquid water absorption and an intensity
offset-correction reduced uncertainties in $NO_2$ SCDs to $0.7 - 0.8 \times 10^{15}$ molec. cm$^{-2}$ (up to $\pm 35$ %). Lorente et al. (2017)
improved the AMF calculation method via the extension of the AMF look-up table with more reference points, and a correction
for the sphericity of the atmosphere. The ancillary data for the AMF calcultion has also improved relative to earlier algorithms
such as DOMINO v2 (Boersma et al., 2011): surface albedo from the 5-year OMI albedo climatology (Kleipool et al., 2008),
cloud information from the improved OMI $O_2$-$O_2$ algorithm (Veefkind et al., 2016), and a priori $NO_2$ profiles from TM5-MP at
$1° \times 1°$ (Williams et al., 2017). The study by Lorente et al. (2017) also showed that substantial differences between AMFs arise





when different a priori $NO_2$ profiles (as well as surface albedo and cloud properties) are used in the retrieval. This underlines that a re-calculation of the tropospheric AMFs based on simulated WRF-Chem $20 \times 20$ km$^2$, replacing the coarse TM5-MP $1° \times 1°$ $NO_2$ profiles, may help to reduce model-satellite differences (Lamsal et al., 2010; Vinken et al., 2014b), and we will explore this further below.

## 2.4 AMF re-calculation

We take care to remove inconsistencies in the model-satellite comparison introduced by different assumptions about the vertical $NO_2$ profile in the satellite product compared to the model. The AMF calculation requires assumptions about the vertical profile of $NO_2$ to convert slant columns into vertical columns. We replace the a priori TM5-MP $NO_2$ profiles (at $1° \times 1°$) by WRF-Chem $NO_2$ profiles at a $20 \times 20$ km$^2$ resolution. This has two advantages: 1) model-satellite comparisons are no longer affected by differences in model assumptions between WRF-Chem and TM5-MP that lead to different vertical $NO_2$ profiles, and 2) the higher resolution WRF-Chem setup resolves spatial gradients in the a priori profile that are not appropriately captured in TM5-MP due to the coarser model resolution. Single-orbit results indicate that re-calculation of the AMFs leads to retrieved columns that are $1 \times 10^{15}$ molec. cm$^{-2}$ higher in densely populated areas, and lower or unaffected in surrounding non-urban regions. This effect has been seen before in earlier studies (Huijnen et al., 2010; Heckel et al., 2011; Russell et al., 2011; Maasakkers, 2013; Vinken et al., 2014b).

We apply the method described by Lamsal et al. (2010) and Boersma et al. (2016) to replace the TM5-MP vertical $NO_2$ profile by the WRF-Chem profile in the calculation of the air mass factor (AMF):

$$M_{trop,WRF-Chem} = M_{trop,TM5} \times \frac{\sum_{l=1}^{L} A_{trop,l} x_{l,WRF-Chem}}{\sum_{l=1}^{L} x_{l,WRF-Chem}} \qquad (1)$$

where $M_{trop}$ is the tropospheric AMF based on an assumed profile from WRF-Chem or TM5, $A_{trop,l}$ is the tropospheric averaging kernel element for layer $l$, $x_{l,WRF-Chem}$ is the $NO_2$ column density in model layer $l$, and $L$ is the uppermost TM5-MP layer in the troposphere. The tropospheric averaging kernel in Eq. 1 is defined as follows (Boersma et al., 2017b): $A_{trop} = A \times \frac{M}{M_{trop}}$, where $M$ and $M_{trop}$ refer to the AMF and the tropospheric AMF, respectively. Note that the WRF-Chem vertical $NO_2$ profile has been sampled at the TM5-MP vertical layer structure, so $l$ refers to TM5-MP model layers.

## 3 Top-down NO$_x$ emissions: methods

Satellite-detected $NO_2$ columns are sensitive to $NO_x$ emissions at the surface. We exploit this dependence to derive satellite-based surface $NO_x$ emissions using local OMI $NO_2$ columns. We apply an improved version of the mass balance procedure (Martin et al., 2003; Lamsal et al., 2011; Vinken et al., 2014b), which accounts for non-linear feedback from $NO_x$ emission changes on $NO_2$ concentrations via OH:

$$E_{td} = E_{bu} \left( 1 + \beta(1 + \gamma) \frac{C_{OMI,bu} - C_{WC,bu}}{C_{WC,bu}} \right) \qquad (2)$$





where $E_{bu}$ and $E_{td}$ represent $NO_x$ emissions from the bottom-up inventory ($bu$) and the satellite-based top-down estimate ($td$), respectively. $C_{WC,bu}$ represents the monthly-averaged $NO_2$ vertical column density (VCD) simulated by WRF-Chem, and $C_{OMI,bu}$ is the monthly averaged modified QA4ECV OMI $NO_2$ VCD using air mass factors based on the original WRF-Chem $NO_2$ vertical profile ($C_{WC,bu}$, see Section 2.4). WRF-Chem $NO_2$ VCDs are co-sampled with valid OMI observations. We only use OMI and WRF-Chem data for pixels with valid satellite observations for at least 4 days in July 2015 to minimize the random error in the satellite retrieval.

We account for the nonlinear $NO_x$-OH chemistry feedback via a dimensionless scaling factor $\beta$, for which we performed a perturbation simulation with surface emissions increased by 20%:

$$\beta = \frac{\Delta E_{bu,1.2}/E_{bu}}{\Delta C_{bu,1.2}/C_{bu}} = \frac{0.2 C_{bu}}{\Delta C_{bu,1.2}} \tag{3}$$

where $C_{bu}$ are the $NO_2$ columns after a WRF-Chem simulation with bottom-up $NO_x$ emissions, and $\Delta C_{bu,1.2}$ is the change in $NO_2$ columns after perturbing bottom-up $NO_x$ emissions by +20%. In low-$NO_x$ environments, this perturbation leads to higher OH levels and thus to more efficient $NO_x$ loss to $HNO_3$, so that a $\beta > 1$ is needed to achieve column agreement. In $NO_x$-rich environments, however, OH levels are suppressed by enhanced $NO_x$ emissions so that the relative increase in $NO_2$ columns is larger than 20%, resulting in a $\beta < 1$. The use of $\beta$ to account for the sensitivity of the $NO_2$ column to local emissions is essentially a linearization step of non-linear effects due to chemistry.

Application of Equations 2 and 3 would lead to updated $NO_x$ emissions, and consequently also to modifications in the WRF-Chem $NO_2$ profile shapes in response to the updates (e.g. Vinken et al., 2014b). This is accounted for via $\gamma$, which we also obtain from the simulation with +20% perturbed emissions:

$$\gamma = \frac{(C_{OMI,1.2} - C_{OMI,bu})/C_{OMI,bu}}{(C_{WC,1.2} - C_{WC,bu})/C_{WC,bu}} \tag{4}$$

where $C_{WC}$ represents the WRF-Chem $NO_2$ vertical column density (VCD), and $C_{OMI}$ represent the OMI $NO_2$ VCD retrieved using WRF-Chem $NO_2$ vertical profiles from the bottom-up simulation ($C_{WC}$), for the bottom-up (subscript $bu$) and emission perturbation simulation (subscript 1.2), respectively. Our approach to calculate $\gamma$ differs from Vinken et al. (2014b), who derived $\gamma$ from a separate simulation after accounting for $\beta$. Our approach requires one less forward simulation and is thus computationally more efficient, with little impact (<3%) on total derived emissions compared to the approach by Vinken et al. (2014b).

We calculate the scaling factors $\beta$ and $\gamma$ for all land-based and shipping lane WRF-Chem cells based on monthly mean $NO_2$ columns (i.e., ocean-based pixels with emissions above a threshold value of 1 mol km$^{-2}$ h$^{-1}$). These pixels thus also include shipping lanes and offshore oil platforms. OMI-inferred emission changes are calculated locally, i.e. for each individual model cell for which the aforementioned data availability criteria are fulfilled. This differs from previous work where these factors were calculated for regions containing multiple model cells (Vinken et al., 2014a, b) or for individual pixels in global models with a coarse resolution (e.g. Lamsal et al., 2011).



We discard the effect of transport of $NO_2$ away from the source region ('smearing'). In July, solar intensity in Europe is close to its annual peak, which means that the $NO_2$ lifetime is short due to efficient oxidation. Therefore, the clear-sky monthly mean $NO_2$ column difference between model and satellite is indicative of local $NO_x$ emission updates. Previous studies showed that this method reduces the model-satellite $NO_2$ column difference but does not resolve it completely (e.g. Vinken et al., 2014b;

Ghude et al., 2013) as a result of the linearization that is applied in the perturbation calculation. Nonetheless, we will show in this study that the systematic bias between WRF-Chem and OMI $NO_2$ columns is largely removed after application of Eqns. 2-4.

## 4 Bottom-up model evaluation

### 4.1 Surface $O_3$

We start our evaluation of $O_3$ chemistry in WRF-Chem (with bottom-up $NO_x$ emissions, i.e. not yet based on the OMI-inferred $NO_x$ emissions) by a comparison of monthly-averaged, 24-hour mean surface ozone simulations with AirBase observations (Fig. 1, panels a and b, and Table 1). WRF-Chem reproduces the spatial distribution of surface ozone satisfactorily, with an increase in surface $O_3$ concentrations from north to south, as reported elsewhere (e.g. Mar et al., 2016). Highest concentrations are found around the Mediterranean basin. $O_3$ concentrations over Central and Southern Europe are underestimated in WRF-

Chem. Simulated monthly-averaged concentrations do not exceed 110 µg m$^{-3}$, while higher concentrations were observed at several stations in the southern part of the domain. Most notably, WRF-Chem does not capture observed high concentrations of ±130 µg m$^{-3}$ in northern Italy. The good agreement between WRF-Chem and in situ data in the western part of the domain close to the model boundaries with a prevailing westerly circulation indicates that the model boundary conditions describe inflow of long-lived compounds such as $O_3$ from the western boundary well.

Monthly averaged ozone concentrations are an important and widely used metric to evaluate model skill, but are not necessarily indicative of the peak ozone concentrations that typically occur in the afternoon. These monthly averages include the nocturnal conditions with generally the presence of stable boundary layers, in which the titration of ozone in the $NO_x$-saturated regions is difficult to model (e.g. Im et al., 2015). The simulated and observed monthly averaged ozone concentrations at 12:00 h UTC (Fig. 1, panels c and d) demonstrate a similar geographical distribution compared to the monthly average, but with

higher values because photochemical ozone production generally peaks during daytime. This figure demonstrates that peak ozone values occur around the Mediterranean basin, most prominently in North Italy and Spain, where the levels of sunlight and ozone precursor concentrations are high. WRF-Chem shows elevated ozone with respect to adjacent areas, but maximum simulated ozone levels do not exceed 120 µg m$^{-3}$. This underestimation of peak ozone concentrations is also apparent from in Fig. 8b (discussed in more detail in Sect. 6), which shows the simulated versus the observed 12:00 h UTC ozone concentrations.

Our results are in agreement with previous regional chemistry model evaluations for Europe. Such studies typically focus on seasonal variability; we compare our results with the results for European summer (JJA) from those studies. Im et al. (2015) found that a model ensemble underestimates the daytime maximum $O_3$ concentration for sites where observed $O_3$ concentrations exceed 120-140 µg m$^{-3}$, which agrees with our results. In that study, the ensemble mean model bias tends



to become more negative for observed concentrations above 80 µg m$^{-3}$ (Im et al., 2015). The two ensemble members that use CBM-Z chemistry, similar to our WRF-Chem model set-up, are qualitatively in line with the ensemble mean, lending support to the use of CBM-Z in this study. Mar et al. (2016) compared two chemical mechanisms in a WRF-Chem evaluation study over Europe and reported large differences in the representation of peak summer (JJA) ozone: one chemistry model

(MOZART) overestimates mean and MDA8 ozone, while simulations with the other chemistry scheme (RADM2) shows underestimations of peak ozone that are in line with our findings. We will discuss the dependence of ozone simulation on the chemical mechanism choice in detail in Sect. 7. The ensemble model mean daytime ozone concentration in Solazzo et al. (2012) is underestimated by 10-30 µg m$^{-3}$ in four sub-regions of the European continent. Tuccella et al. (2012) analyzed WRF-Chem $O_3$ concentrations for 2007 and found that yearly-averaged mid-day ozone is underestimated by approximately 10 µg m$^{-3}$. The

model performance in the aforementioned studies is qualitatively similar to our findings and the magnitude compares well. Overall, most studies consistently show underestimated daytime $O_3$, regardless of the chemical mechanism, model resolution and other model assumptions. To further explore the potential role of a model misrepresentation of $NO_2$ concentrations in explaining this model $O_3$ bias, the next sections will focus on a model comparison with in situ and remote sensing data for $NO_2$.

### 4.2  Surface $NO_2$

Fig. 2 a and b present a comparison of monthly-averaged surface concentrations of $NO_2$ between WRF-Chem and AirBase (note the logarithmic scale). Performance statistics are shown in Table 1. We find that WRF-Chem reproduces the spatial distribution well, with peak $NO_2$ occurring in Northwest Europe and North Italy. In these regions with high $NO_x$ emissions, average WRF-Chem-simulated concentrations are however underestimated by up to 10 µg m$^{-3}$ compared to observations. AirBase con-

centrations show a region with elevated $NO_2$ concentrations in Southwest Germany. WRF-Chem also shows elevated $NO_2$ concentrations in this region, but does not reach such elevated concentrations. Overall, WRF-Chem shows more spatial heterogeneity in surface $NO_2$ concentrations than is apparent from the observations. Observed $NO_2$ concentrations in background areas in Spain, France and Eastern Europe are 2-5 µg m$^{-3}$ or higher, while the model consistently simulates values <2 µg m$^{-3}$ in these regions. This overall underestimation is also seen in Fig. 8, where the simulated daily mean $NO_2$ concentration is shown

against AirBase observations. The model performance of our WRF-Chem setup is in line with previous WRF-Chem studies. Mar et al. (2016) found small overestimations (0.67-2.96 µg m$^{-3}$) in mean $NO_2$. Another study found an annual average mean bias of -0.9 µg m$^{-3}$, caused by underestimations of peak $NO_2$ in WRF-Chem (Tuccella et al., 2012).

A comparison between WRF-Chem and AirBase monthly-averaged 12:00 h UTC $NO_2$ concentrations is presented in Figure 2c and d and Table 1. We find that WRF-Chem on average strongly underestimates mid-day $NO_2$ concentrations by 2.96 µg

m$^{-3}$ (38.5%).

### 4.3  $NO_2$ VCD

Before we perform a comparison between $NO_2$ VCDs from WRF-Chem and OMI, we first discuss the effect of the $NO_2$ profile shape on the OMI-retrieved columns. Figure 3 shows the change in the monthly-averaged OMI $NO_2$ column density





after replacing TM5-MP $NO_2$ profiles by WRF-Chem profiles using the procedure described in Sect. 2.4. The OMI $NO_2$ VCDs change most prominently over urban/industrial areas such as the Netherlands, Paris, Berlin, Madrid, Milano and Rome. The background areas are largely unaffected, or show small ($\pm$ 0.2 $\times 10^{15}$ molec. cm$^{-2}$) $NO_2$ VCD increases (e.g. Spain) or decreases (regions in France, Germany, Poland, Ukraine and Romania). The vertical $NO_2$ profile over sea regions in western

Europe strongly peaks at the surface, because shipping $NO_x$ in WRF-Chem is emitted in the lowermost model layer. Overall, the average $NO_2$ column change over non-land regions is small (<2%).

We subsequently compare WRF-Chem to this modified OMI product. The monthly-averaged $NO_2$ vertical column densities from WRF-Chem and OMI are displayed in Fig. 4. The model is sampled at 12:00 h UTC, close to the OMI overpass time of $\pm$13:30 h LT, and is co-sampled with valid satellite observations. There is good agreement in the spatial distribution of

monthly-averaged $NO_2$ VCDs (r$^2$ = 0.68). $NO_2$ columns are underestimated by $0.3\times10^{15}$ molec. cm$^{-2}$ on average, with strong underestimations of up to $2\times10^{15}$ molec. cm$^{-2}$ in urban and industrial northwestern Europe. WRF-Chem overestimates $NO_2$ columns in some isolated urban areas with high $NO_x$ emissions such as London, Madrid, Rome, and in parts of Eastern Europe.

We note that Fig. 4 shows small underestimations of the simulated $NO_2$ VCD compared to OMI ($\pm0.2 \times 10^{15}$ molec. cm$^{-2}$) in background regions (e.g. the Alps, rural Spain and France, Scandinavia) and over the oceans. Simulated $NO_2$ columns

therefore show stronger spatial gradients than OMI-retrieved columns, which is in line with Huijnen et al. (2010). Other distinct underestimations in the simulated $NO_2$ columns compared to OMI indicate a misrepresentation of emissions. For example, the region in Northern Spain where WRF-Chem underestimates the $NO_2$ column compared to OMI observations by $2 \times10^{15}$ molec. cm$^{-2}$ suggests that emissions from power plants in this region are still strong in July 2015, despite previously reported reductions (Zhou et al., 2012).

We have shown that our WRF-Chem set-up with bottom-up emissions underestimates $NO_2$ with respect to both surface and column measurements. To combine these model comparisons against different data sources, we already discuss parts of Fig. 9, which compares the agreement between simulations with bottom-up and top-down emissions. Fig. 9a shows the relative difference of WRF-Chem against AirBase and OMI $NO_2$ binned as a function of bottom-up anthropogenic emission strength. This shows an overall underestimation of WRF-Chem at the surface and in the troposphere, except for regions with

strongest emissions. There is a relatively larger model underestimation of surface $NO_2$ than of the $NO_2$ VCD in regions with low emissions, suggesting that emissions are generally too low in WRF-Chem, but especially that emissions in rural background regions, are underestimated. This, in turn, suggests that the representation of surface $NO_x$ emissions in WRF-Chem (anthropogenic emissions for 2011 and on-line calculated natural soil emissions) are too low to explain the observations in July 2015. In the following section, we will derive satellite-constrained $NO_x$ emissions and discuss potential reasons for this

mismatch.



## 5 Satellite-derived NO$_x$ emissions

### 5.1 Top-down emissions

We derive top-down NO$_x$ emissions using the method described in Section 3. Fig. 5 shows the July total bottom-up and top-down surface NO$_x$ emissions and their difference. Top-down NO$_x$ emissions amount to 498 Gg N, which is 56% higher than the bottom-up inventory, and increases occur across the domain (Fig. 5c). NO$_x$ emissions are reduced in several isolated grid cells that generally correspond to urban areas. The difference between top-down and bottom-up emissions is larger than the 16% increase reported by Miyazaki et al. (2017), although that study found strong (40-67%) local increases in areas with high NO$_x$ emissions such as Belgium, western Germany and northern Italy.

Our top-down emissions are much higher than the bottom-up emissions over Germany and Poland. Over Belgium and the Netherlands, the difference between top-down and bottom-up emissions is also substantial, but notably smaller despite larger differences between OMI and WRF-Chem NO$_2$ columns over the low-countries (Fig. 4c). This reflects the chemical regime with very high bottom-up NO$_x$ emissions in this region, resulting in suppressed mid-day OH concentrations, and consequently, longer NO$_2$ lifetimes (as diagnosed by low beta values over northwestern Europe in Supp. Fig. 1).

We subsequently replace bottom-up emissions with our observation-constrained top-down NO$_x$ emissions and perform a new WRF-Chem simulation. As expected, the new NO$_2$ columns agree much better with the OMI NO$_2$ columns than those from the simulation with bottom-up emissions (Fig. 6). WRF-Chem with bottom-up emissions generally underestimates OMI NO$_2$ columns by 23.4%. As expected, the simulations with the top-down emissions agree better with OMI, and the slope of 0.98 between the new WRF-Chem and OMI NO$_2$ columns (Fig. 6b) suggests that the systematic underestimation in the model is effectively resolved by applying the top-down emissions. The mean relative error is reduced to -7.5%, and the spatial correlation coefficient between WRF-Chem and OMI NO$_2$ also improves considerably (from 0.68 to 0.84).

### 5.2 Attribution to emission sources

Fig. 7 shows the bottom-up and top-down NO$_x$ emissions as a function of the bottom-up anthropogenic emission strength. This comparison demonstrates that top-down NO$_x$ emissions are higher than bottom-up emissions regardless of the emission strength. However, top-down emissions are 50-100% higher than bottom-up estimates for relatively weak emissions between 0.5-50 Mg N month$^{-1}$ cell$^{-1}$, and only up to 20% higher for some urban and industrial hotspots (Fig. 7b). This 0.5-50 Mg N month$^{-1}$ range is dominated by WRF-Chem grid cells located in the rural areas of Europe, excluding the largest urban agglomerations as well as low-emission regions such as mountainous areas. Our substantially larger top-down emissions partly reflect a required increase in NO$_x$ emissions in areas where soil NO emissions are expected to be a dominant NO$_x$ source. Soil NO emissions are simulated in WRF-Chem using an implementation of the MEGAN biogenic emission model. The observed discrepancy between the WRF-Chem-simulated and OMI-observed NO$_2$ VCD triggers to assess how much of this discrepancy can be attributed to this model's representation of soil NO emissions.

To separate the soil NO$_x$ contribution from the anthropogenic emission updates, we perform a simple budget calculation as a first-order constraint on the partitioning of the top-down emissions between their anthropogenic and soil-based sources. We



assume that the relative difference in anthropogenic sources is uniform over the emission bins in Fig. 7. This factor is calculated as the median of the relative change in emissions for the three highest bins (>50 Mg N cell$^{-1}$ for July, see Fig. 7), and amounts to 0.22. This allows us to attribute the remaining emission difference to soils. Based on this crude first estimate, we derive top-down soil $NO_x$ emissions to be 112 Gg N month$^{-1}$, versus WRF-Chem/MEGAN-simulated bottom-up soil NO emissions of

only 15 Gg N month$^{-1}$. The anthropogenic enhancement factor is relatively uncertain, but does not strongly impact our derived posterior soil $NO_x$ emission estimate: if, instead of the median (m = 0.22), we use the mean relative change in emissions for the three highest bins ($\mu = 0.41$), our soil contribution is still a factor >4 larger (69.0 Gg N month$^{-1}$) compared to WRF-Chem's simulated bottom-up soil NO source. Therefore, this first-order estimation suggests that a substantial fraction (43-69%) of the $NO_x$ emission increment after optimization can be attributed to soils.

To evaluate the derived total soil $NO_x$ emissions, we perform a comparison with literature-based estimates in Table 2. We find that bottom-up soil $NO_x$ emissions are underestimated by a factor 5-7 compared to previous studies. In some of those studies (e.g. Ganzeveld et al., 2010), land use management practices (fertilizer and manure application) provide a substantial contribution to European soil NO emissions, a feature that appears to be missing in the representation of soil NO emissions in WRF-Chem. This supports our hypothesis that a substantial fraction of the increase in surface $NO_x$ emissions may be attributed

to soils. We will discuss this further in Sect. 7.

## 6   Emission scaling impacts on surface $NO_2$ and $O_3$

### 6.1   Nitrogen dioxide

Table 1 summarizes the model performance of our bottom-up and top-down WRF-Chem simulations against a large number of AirBase $NO_2$ observations throughout Europe in July 2015. The simulation with top-down emissions improves upon the a

priori run in all metrics. Most notably, the model index of agreement ($d$) improves by 0.10 (14%). The modified model set-up still slightly underestimates the highest monthly averaged $NO_2$ observations, as indicated by a slope of 0.86. However, the low bias in WRF-Chem surface $NO_2$ concentrations with respect to AirBase improves from -2.5 to -1.1 µg m$^{-3}$.

Compared to the monthly average, we find little improvement in WRF-Chem's skill to predict surface $NO_2$ at 12:00 h UTC. The model's low bias in $NO_2$ reduces from -3.0 to -2.6 µg m$^{-3}$ and the index of agreement improves by only 0.02 (4%). This

more modest improvement in performance can be understood from mid-day surface $NO_2$ concentrations being more strongly driven by photochemical removal processes and boundary layer development than the 24-hour mean $NO_2$ levels, that are more sensitive to $NO_x$ emissions due to strongly reduced mixing and photochemistry at night. Fig. 8 displays WRF-Chem monthly, 24-hour mean $NO_2$ concentrations against AirBase observations, for the bottom-up (black) and top-down (red) simulations. The model orthogonal distance regression (ODR) slope improves considerably, while the explained variance of the model

improves slightly to 0.46.

Fig. 9 shows the relative biases between WRF-Chem and observed $NO_2$ as a function of (binned) bottom-up anthropogenic NO emission strength. Both the WRF-Chem simulations with bottom-up emissions (Fig. 9a) as well as the simulation with top-down emissions (Fig. 9b) show a low bias against OMI and AirBase for regions with low emissions, and a positive relative



bias in regions with stronger emissions. The relative bias is however considerably reduced in the simulation with top-down $NO_x$ emissions, both at the surface and in the column. However, WRF-Chem still displays a stronger relative bias compared to AirBase than compared to OMI. This feature can likely be attributed to a difference in spatial scales between the $20 \times 20$ km²-resolution model versus the footprint area of local AirBase measurements, which can be easily influenced by a nearby $NO_x$

source that is less well captured in the model, due to instantaneous mixing over a larger volume. Another potential explanation for lower relative bias of WRF-Chem compared to AirBase than compared to OMI is interference of in situ measurements with molybdenum converters (see Sect. 2.2). This is in line with our previous finding that the slope of the top-down $NO_2$ column regression fit approaches 1, while the slope of the fit for in situ $NO_2$ observations is still below 1. We also note that the spread in the relative bias compared to AirBase increased for the top-down simulation, with more positive relative bias values for

all bins. Nonetheless, the results shown in Fig. 9 provide confidence regarding application of the model as a tool to reconcile local-scale bottom-up emissions and concentrations with larger-scale remote sensing-based $NO_2$ measurements.

## 6.2   Ozone

Next, we address our main question whether the improved simulation of $NO_2$ leads to better model performance for surface ozone simulations. We find that WRF-Chem with top-down emissions improves upon the bottom-up simulation for both the

24-hour mean, as well as the 12:00 h UTC and MDA8 ozone metrics. The model index of agreement improves by 0.08-0.11 (13-17 %, Table 1). However, the top-down model still simulates too low surface $O_3$, especially over southern, eastern and central Europe, where observed surface $O_3$ exceeds 80 μg m⁻³ at 12:00 h UTC (see Fig. 10).

A comparison between monthly averaged mid-day $O_3$ concentrations from the bottom-up and top-down simulation (Fig. 10, panels a and b, respectively) shows that ozone increases across the model domain. This particularly improves the WRF-

Chem-AirBase agreement in large parts of western and Central Europe. The simulated ozone values in northern Italy remain underestimated.

Surface ozone concentrations display a strong increase due to the use of top-down $NO_x$ emissions (Fig. 10). The areas where ozone concentrations increase by >10 μg m⁻³ largely coincide with regions where top-down $NO_x$ emissions are much higher than the bottom-up emissions (Fig. 5c), such as in northern Spain, southern Germany, southern Poland, Croatia, Serbia, western

Greece and southern Romania. There are also strong simulated ozone increases in central France and over the Adriatic Sea. These regions are all characterized as (rural) background areas, where ozone formation is strongly sensitive to the increases introduced in the $NO_x$ emissions for the relatively low bottom-up anthropogenic and soil emissions. We find decreases in ozone around the main shipping lanes, where the higher $NO_x$ emissions further enhance ozone titration. The enhanced titration also reduced simulated surface ozone around urban regions such as Barcelona, Rome, and Paris. The increases in surface

$NO_x$ emissions in the BeNeLux and western Germany slightly increase simulated mid-day surface ozone. Ozone production is less sensitive to $NO_x$ emissions in these high $NO_x$-emitting regions compared to the unpolluted background (Beekmann and Vautard, 2010; Mar et al., 2016; Jin et al., 2017).

Fig. 8 shows that $O_3$ simulations with the higher top-down $NO_x$ emissions lead to a somewhat better match between modeled and observed surface $O_3$, with an improvement in spatial correlation coefficient from 0.43 to 0.57, and an increase in slope from



0.33 to 0.40. Overall, the model low bias has reduced from -15 to -8 μg m$^{-3}$, which indicates that the use of OMI NO$_2$ VCD data to constrain WRF-Chem surface NO$_x$ emissions results in a considerable improvement regarding simulation of surface layer O$_3$ concentrations.

## 7    Discussion

In this study we demonstrate the added value of deriving satellite-based NO$_x$ emissions in (regional) air pollution models for simulations of summertime ozone, focusing on July 2015 over Europe. We use a modified version of the mass balance approach introduced by Martin et al. (2003), with further improvements by Lamsal et al. (2011) and Vinken et al. (2014b). Although many studies report differences in simulated (surface) ozone concentrations after applying this mass balance approach (e.g. Ghude et al., 2013), we are aware of only one other study that used observations to validate subsequent ozone changes. Verstraeten et al. (2015) used TES O$_3$ observations in the global chemistry model TM5 in a study on trans-continental transport of Asian air pollution, and found an improved model-satellite agreement in lower-tropospheric ozone. However, their approach did not allow for an evaluation of model performance closer to the surface.

The mass balance approach that we used to derive observation-constrained European NO$_x$ emissions has several important advantages over more formal inversion methods that are applied in the literature (e.g. Miyazaki et al., 2014, 2017). The method is highly traceable due to the simple calculation of scaling parameters from model output for a baseline and perturbation simulation, and column NO$_2$ measurements. However, the linearization (see Sect. 3) oversimplifies the nonlinearity of the NO$_x$-O$_3$ chemistry, which means that the model-satellite discrepancy is not resolved completely after one iteration. Additionally, the approach is only applicable on a pixel-basis when the NO$_x$ lifetime is sufficiently short to discard the contribution of transport from adjacent model NO$_2$ columns. The model-satellite difference for a simulation we performed for March 2015 (not shown) shows less spatial heterogeneity over regions with a diffuse spatial distribution of NO$_x$ sources (e.g. Germany). These shortcomings can be resolved by averaging the signal over multiple grid cells, or by applying more formal inversion methods.

Our results demonstrate that surface NO$_x$ emissions in our WRF-Chem configuration are increased substantially after applying an emission scaling approach. In a first-order budget calculation we derive that 43-69% of this total increase can be attributed to soil NO$_x$. This is diagnosed from the notably higher relative increase in emissions in regions with moderate anthropogenic emissions compared to regions with low and high anthropogenic emissions. We therefore conclude that the contribution of soil NO$_x$ to total surface emissions is likely underestimated in our model set-up. Additionally, our top-down soil NO$_x$ emission estimate, derived with a budget calculation, agrees well with previous estimates for European summer (Table 2). Our findings are in line with a previous study (Oikawa et al., 2015) that, using WRF-Chem with MEGAN soil NO$_x$ emissions, found a strong underestimation of NO$_x$ emissions in a high-temperature agricultural region.

The comparison against in situ NO$_2$ observations from the AirBase network may be hindered by interference of reactive N species for measurements with molybdenum converters. The type of converter is not reported in the database. Literature-reported estimates of measurement overestimations due to this interference are 22% (Dunlea et al., 2007) and 5-18% (Boersma





et al., 2009) at urban sites, and 20-42% at a rural site (Steinbacher et al., 2007). A correction factor can be applied to obtain corrected $NO_2$ measurements from observations using a molybdenum converter, which is on average 0.4-0.6 in summer, but with a large spread (0.2-0.8) (Lamsal et al., 2008, 2010). The strongest corrections of molybdenum-based in situ $NO_2$ measurements are needed in remote environments, where $NO_x$ is a relatively smaller component of the total reactive nitrogen budget

compared to areas closer to $NO_x$ sources (Lamsal et al., 2008). We hypothesize that this can partially explain the remaining model-observation mismatch for $NO_2$ after the use of top-down emissions.

Despite the demonstrated improvement in ozone simulations, our simulation with OMI-derived top-down $NO_x$ emissions still misrepresents the high tail of the ozone distribution. We believe that there is a potential explanatory role for local to regional meteorological processes. The representation of several mesoscale phenomena requires a higher model resolution

than $20 \times 20$ km$^2$. For example, Millán et al. (1997) demonstrated that local re-circulation of residual air masses from higher aloft, containing elevated $O_3$ transported aloft during previous days, can be entrained in the boundary layer and contribute substantially to air pollution episodes in southern Europe. This is supported by an analysis of measured ozone (precursors) in northeast Spain by Querol et al. (2017), where this mesoscale circulation pattern was found to contribute to concentrations that exceed the information threshold value set by the European Union (180 µg m$^{-3}$), alongside contributions from locally emitted

$NO_x$ and biogenic VOCs.

Simulations of surface ozone in AQ models are also impacted by the choice of chemical parameterization. Recently, several studies have investigated the influence of the chemical mechanism on simulated $NO_x$ and $O_3$ concentrations. Regarding ozone chemistry, chemical mechanisms differ predominantly in two aspects: 1) the grouping of VOC species in species categories ("lumping") according to their chemical structure or number of C-atoms, and 2) the inorganic rate coefficients involved in the

catalytic cycling of $NO_x$, $HO_x$ and $O_x$. Especially the latter aspect has a strong influence on simulated $NO_2$ concentrations, and can therefore influence the derivation of top-down emission estimates using satellite observations (Stavrakou et al., 2013). Coates et al. (2016) investigated the maximum ozone formation potential in different chemical mechanisms and found that mechanisms with lumped VOC categories led to lower ozone mixing ratios compared to a mechanism with a near-explicit treatment of VOCs. Knote et al. (2015) found small differences in inorganic rate constants among mechanisms and thus

concluded that VOC representation was the dominating source of uncertainty among mechanisms. However, Mar et al. (2016) performed a WRF-Chem sensitivity study where MOZART inorganic rate constants were applied within RADM2, leading to mean $O_3$ concentration differences of 8 µg m$^{-3}$ between those mechanisms.

In order to test the importance of inorganic $NO_x$-$HO_x$-$O_x$ reaction rates for ozone formation, we implemented inorganic rate constants from three different mechanisms (CBM-Z, RADM2 and MOZART) in a mixed layer model with simplified chemistry

(Janssen et al., 2012). Further details are given in Sect. 5 of the Supplement. Our analysis shows that varying the temperature-dependent rate constant of $HNO_3$ formation ($k_{NO_2 + OH}$) can lead to a spread of 2 ppb for end-of-afternoon ozone values on a typical summer day in a polluted boundary layer. CBM-Z uses the lowest $k_{NO_2 + OH}$ among the considered mechanisms, and thus leads to a higher $NO_2$ lifetime and more $O_3$ formation than in other mechanisms. Therefore, we conclude that modification of inorganic reaction rate constants has a modest effect on simulated $O_3$, but is not likely to lead to increases in simulated $O_3$





in our WRF-Chem configuration. Nevertheless, the model representation of ozone chemistry should be carefully considered in $NO_x$ and $O_3$ air quality studies, besides the representation of $NO_x$ emissions.

Several studies have considered the resolution dependence of air quality simulations. This is especially relevant for $NO_2$, since $NO_x$ emissions display strong variation on the $20 \times 20$ km$^2$ scale applied in this study. Increasing model resolution leads

to better representation of these local gradients and therefore improves simulations of $NO_2$ concentrations (Schaap et al., 2015). Valin et al. (2011) found that an accurate representation of mid-day $NO_2$ columns from highly localized sources requires a high model resolution, but regions with more diffuse sources can be simulated at a coarser resolution of $\pm 10 \times 10$ km$^2$. Although ozone production regimes do not strongly depend on the model resolution in regional models, high resolution models perform better at simulating local $O_3$ titration in freshly emitted NO plumes (Cohan et al., 2006).

Besides the representation of meteorological processes, there is an additional uncertainty related to surface-atmosphere exchange of pollutants. Dry deposition constitutes 17% of the tropospheric sink of ozone, and is the second most important removal process after chemical removal (Hu et al., 2017). Several studies have recently investigated the role of meteorological drivers that determine ozone removal at the surface. However, these meteorological controls are oversimplified in deposition parameterizations. The vapour pressure deficit strongly controls stomatal uptake of ozone, thereby affecting surface ozone

levels in spring to summer in the United States (Kavassalis and Murphy, 2017). Analysis of 10-year $O_3$ flux observations in the northeastern United States revealed that the removal of ozone by the land surface exhibits a strong inter-annual variability, which is not captured in dry deposition parameterizations (Clifton et al., 2017). Lastly, the role of soil moisture has been proposed as a regulator of surface ozone uptake (Tawfik and Steiner, 2013) and is often neglected in parameterizations of dry deposition, even though a recent study found that it can significantly reduce simulated ozone uptake (Anav et al., 2017).

Improving the biophysical representation of the dry deposition process in WRF-Chem will be one of our foci in the future.

Future studies that apply satellite-based constraints on surface $NO_x$ emissions can benefit from observations from the recently launched TROPOMI instrument (Veefkind et al., 2012), which delivers $NO_2$ column data at an unprecedented resolution of $7 \times 3.5$ km$^2$. This has the potential to lead to important improvements in satellite-constrained $NO_x$ emissions. Recent work (Lorente et al., 2019, in review) has applied TROPOMI observations in a column model study to derive emissions from Paris.

The resolution of the instrument additionally enables the focus on more local areas with one dominating source such as soils in agricultural or bare-soil regions.

## 8 Conclusions

We performed a WRF-Chem simulation of $NO_x$ and ozone over Europe for July 2015 and assessed its performance with AirBase in situ observations and OMI $NO_2$ column measurements. We find that WRF-Chem underestimates high surface

ozone concentrations in central and southern Europe, and overestimates lower ozone concentrations in northern Europe. The model also underestimates the spread. The monthly averaged mean bias error (MBE) is relatively small (-2.4 μg m$^{-3}$, 10%). WRF-Chem underestimates daytime increases in ozone concentrations, as evidenced by substantial negative MBE values for the mid-day (12 h UTC) $O_3$ concentration and MDA8 $O_3$ (-15.1 μg m$^{-3}$ and -14.2 μg m$^{-3}$, respectively). We relate the low bias



in surface ozone to biases in ozone precursor concentrations. Of particular relevance are nitrogen oxides, which drive ozone production in much of $NO_x$-limited summertime Europe.

For $NO_2$, we find that WRF-Chem underestimates surface and column $NO_2$ values for most of the domain, with exception of some high-emission regions. With respect to AirBase, WRF-Chem monthly averaged surface $NO_2$ is biased low by -2.5 µg m$^{-3}$

(-73%). The spatial distribution of WRF-Chem column $NO_2$ agrees well with OMI ($r^2 = 0.68$), and a mean underestimation of $0.3 \times 10^{15}$ molec. cm$^{-2}$ (-23%). We attribute the low bias in WRF-Chem $NO_2$ concentrations to underestimations in surface $NO_x$ emissions in WRF-Chem. We subsequently derive optimized $NO_x$ emissions based on the WRF-Chem/OMI relative difference using a mass balance approach. Overall emissions increase from 0.32 to 0.50 Tg N, an increase of 0.18 Tg N (+56%), for July 2015. The updates indicate that $NO_x$ emissions should be scaled up across the domain. The relative increase

in emissions is largest for regions with moderate emission strength (up to 50 Mg N month$^{-1}$ cell$^{-1}$) and coincides with regions where agricultural soil $NO_x$ emissions are substantial. Our optimized soil $NO_x$ emissions amount to 0.1 Tg N, in much better agreement with values from the literature.

A WRF-Chem simulation with optimized $NO_x$ emissions removes the model's systematic bias with respect to OMI $NO_2$, and leads to an improved spatial agreement (slope = 0.98, $r^2 = 0.84$). An evaluation against AirBase $NO_2$ reveals that the

top-down simulation improves particularly in the monthly average, where the systematic mismatch is reduced (slope = 0.89 instead of 0.73) and the mean bias is reduced by 50%. For ozone, the model skill improves particularly for mid-day and MDA8 $O_3$, when local ozone formation occurs and the sensitivity of ozone formation to $NO_x$ concentrations is highest. On average, surface $O_3$ concentrations increase by 6 µg m$^{-3}$ (6%). Still, peak (mid-day) ozone values are underestimated after $NO_x$ emission optimization.

Overall, our findings demonstrate that air quality model simulations combined with in situ and remote sensing observations can be used to infer missing sources of $NO_x$ at the surface. By optimizing $NO_x$ emissions with satellite observations, substantial improvements in simulated ozone can be achieved. Our work shows that this helps to reduce the persistent biases in $O_3$ that most air quality models are suffering from. Projected decreasing trends in anthropogenic $NO_x$ emissions will mean that the contribution of soils to total European $NO_x$ emissions will likely increase in the future, and thus deserves careful attention in

(European) air quality assessments, along with detailed assessments of emissions of volatile organic compounds and wildfires, boundary layer mixing, and chemistry.

*Code and data availability.* WRF-Chem output and re-calculated OMI $NO_2$ columns are available upon request, as well as scripts to re-calculate the tropospheric AMF and the resulting changes in satellite $NO_2$ columns.

*Author contributions.* AV, KFB and LG designed the experiment. AV performed the model simulations and analysis, with support from all

co-authors. AV wrote the manuscript, with contributions from all co-authors.





*Competing interests.* The authors declare no competing interests.

*Acknowledgements.* AV acknowledges funding from NWO's program "Gebruikersondersteuning Ruimteonderzoek" (GO) project ALW-GO/16-17 SASODIE: Space-based assessment of ozone deposition and its impact on ecosystem functioning. The authors acknowledge the free availability of the WRF-Chem model (https://www2.acom.ucar.edu/wrf-chem, last acces: 5 February 2019), in situ data from AirBase (http://discomap.eea.europa.eu/map/fme/AirQualityExport.htm, last access: 14 March 2019), and satellite $NO_2$ column observations from the OMI instrument (http://www.qa4ecv.eu, last access: 12 March 2019). We additionally thank John Paton for his help downloading AirBase in situ measurements.





**Table 1.** Performance statistics of WRF-Chem bottom-up and top-down simulations for July 2015 for several conventionally applied performance metrics (MBE, RMSE, slope and intercept of a linear regression fit of simulations against observations, and $r^2$ from orthogonal distance regression), as well as the index of agreement ( $d = 1 - \frac{\sum_{i=1}^{N}(P_i-O_i)^2}{\sum_{i=1}^{N}(|P_i'|+|O_i'|)^2}$, Willmott, 1982), where $P_i$ and $O_i$ represent simulations and observations, respectively. MBE, RMSE and intercept have unit µg m$^{-3}$, slope, $r^2$ and d are unitless.

| | | Bottom-up | | | | | | Top-down | | | | | |
| | n | MBE | RMSE | slope | intercept | $r^2$ | d | MBE | RMSE | slope | intercept | $r^2$ | d |
|---|---|---|---|---|---|---|---|---|---|---|---|---|---|
| $\overline{[O_3]}$ | 289 | -2.37 | 2.50 | 0.26 | 54.27 | 0.32 | 0.60 | 2.18 | 17.03 | 0.34 | 53.23 | 0.41 | 0.68 |
| $\overline{[O_3]}^{12h}$ | 397 | -15.07 | 24.68 | 0.33 | 51.63 | 0.43 | 0.63 | -7.56 | 19.09 | 0.41 | 51.13 | 0.58 | 0.74 |
| MDA8 $O_3$ | 289 | -14.24 | 24.79 | 0.28 | 55.98 | 0.40 | 0.61 | -7.38 | 19.99 | 0.36 | 55.72 | 0.53 | 0.70 |
| $\overline{[NO_2]}$ | 184 | -2.49 | 3.86 | 0.73 | -0.28 | 0.42 | 0.70 | -1.09 | 3.09 | 0.89 | -0.12 | 0.46 | 0.80 |
| $\overline{[NO_2]}^{12h}$ | 250 | -2.96 | 3.56 | 0.30 | -0.03 | 0.25 | 0.51 | -2.59 | 3.28 | 0.33 | 0.04 | 0.23 | 0.53 |





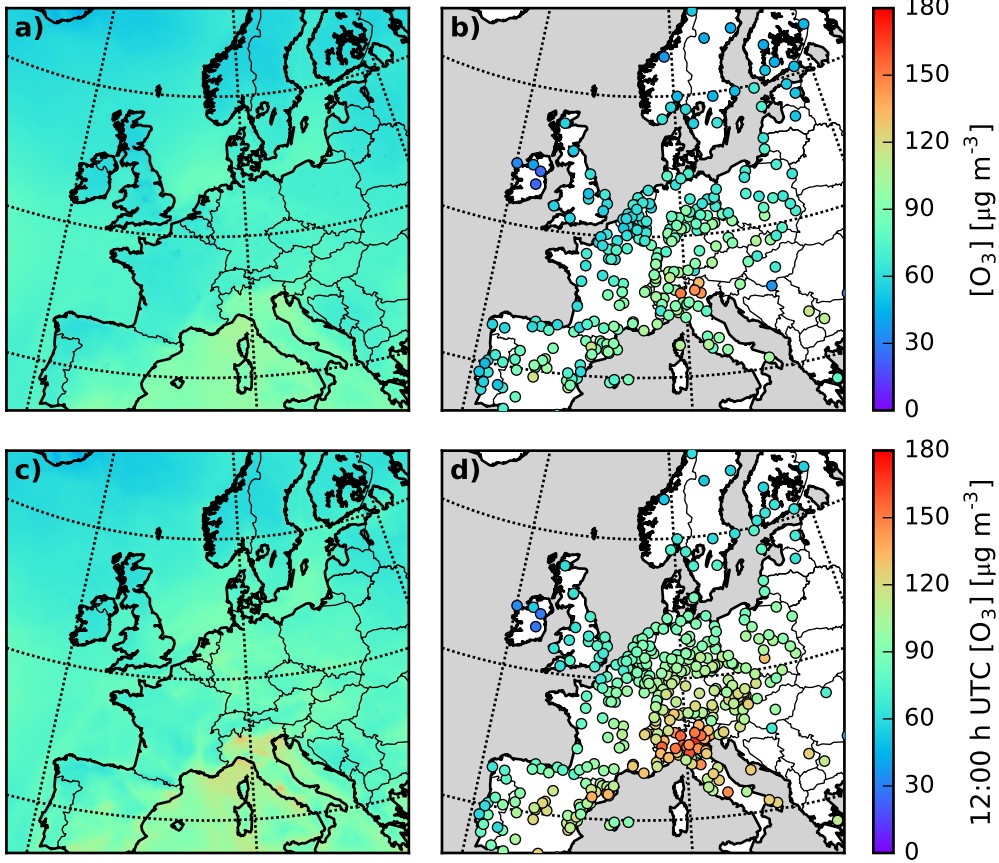

**Figure 1.** Monthly averaged surface $O_3$ and simulated by WRF-Chem with bottom-up $NO_x$ emissions (a & c) and observed at AirBase stations (b & d). Panels a) and b) are monthly averages, and b) and d) are sampled at 12:00 h UTC.





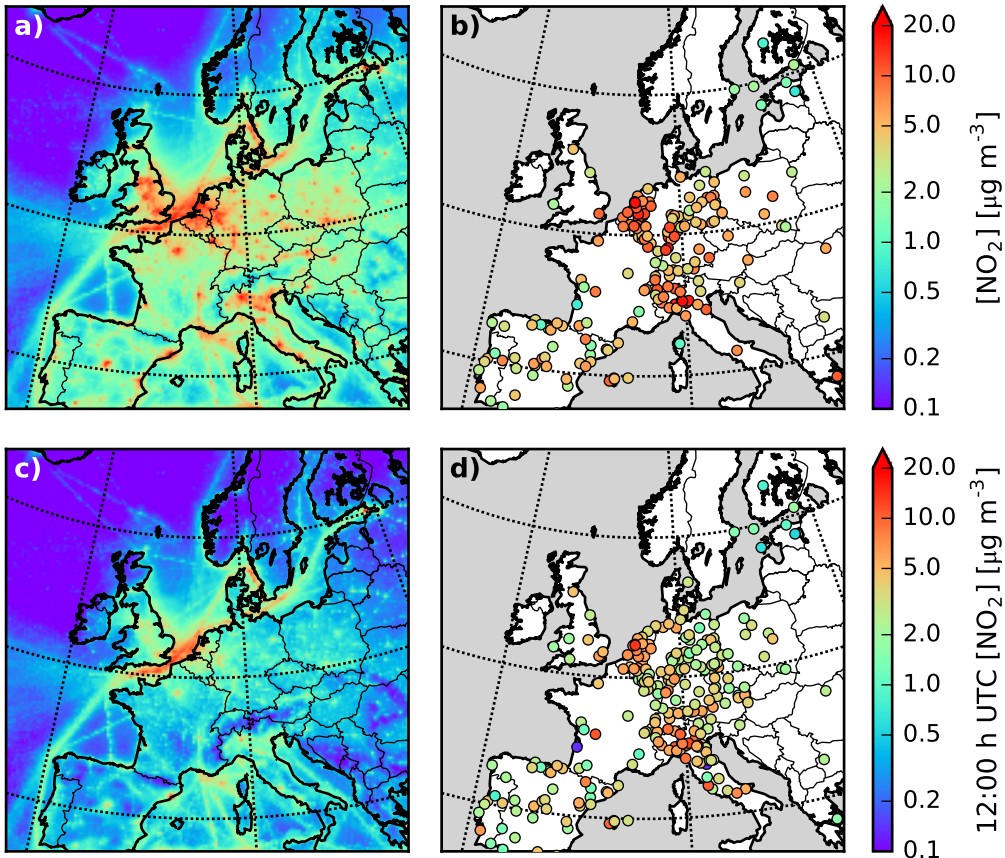

**Figure 2.** As Fig. 1, but for NO₂.

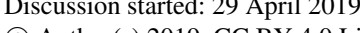



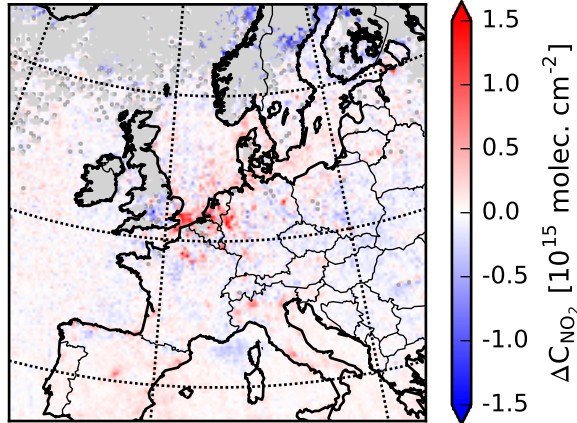

**Figure 3.** Change in monthly-averaged OMI-retrieved NO₂ columns after using WRF-Chem vertical NO₂ profiles to calculate the Air Mass Factors (AMFs) in the OMI retrieval, as described in Sect. 2.4.

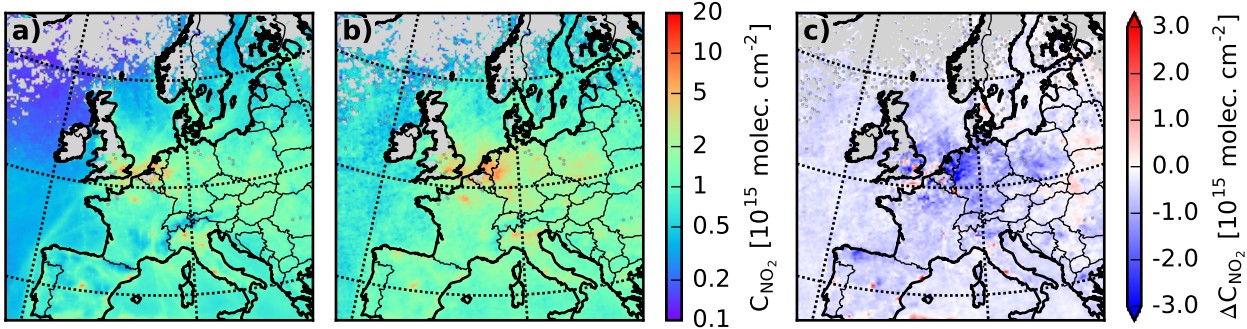

**Figure 4.** Monthly-averaged tropospheric NO₂ vertical column densities from a) WRF-Chem with bottom-up NO$_x$ emissions, b) OMI and c) their difference (WRF-Chem - OMI). WRF-Chem NO₂ columns have been co-sampled with OMI, and pixels are shown when $n_{obs} \geq 4$.





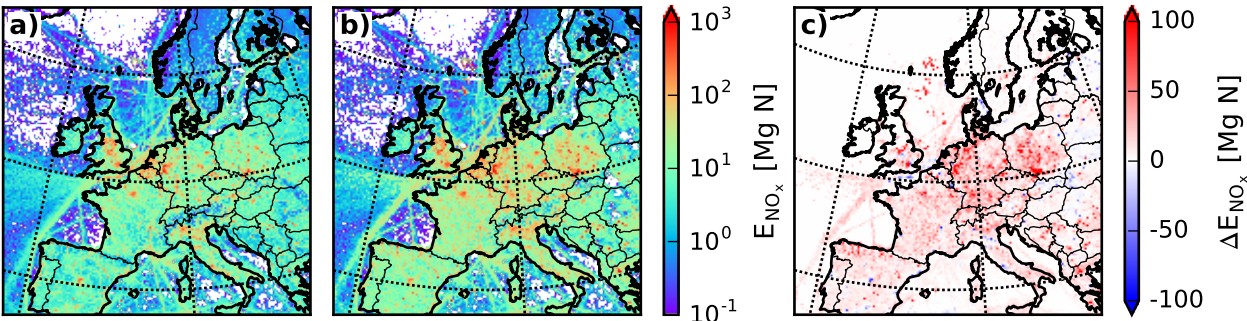

**Figure 5.** Surface NO$_x$ emissions for a) the a bottom-up simulation (TNO-MACC-III anthropogenic + MEGAN soil NO$_x$), and b) the top-down simulation; c) depicts the change in surface NO$_x$ emissions after the recalculation procedure.

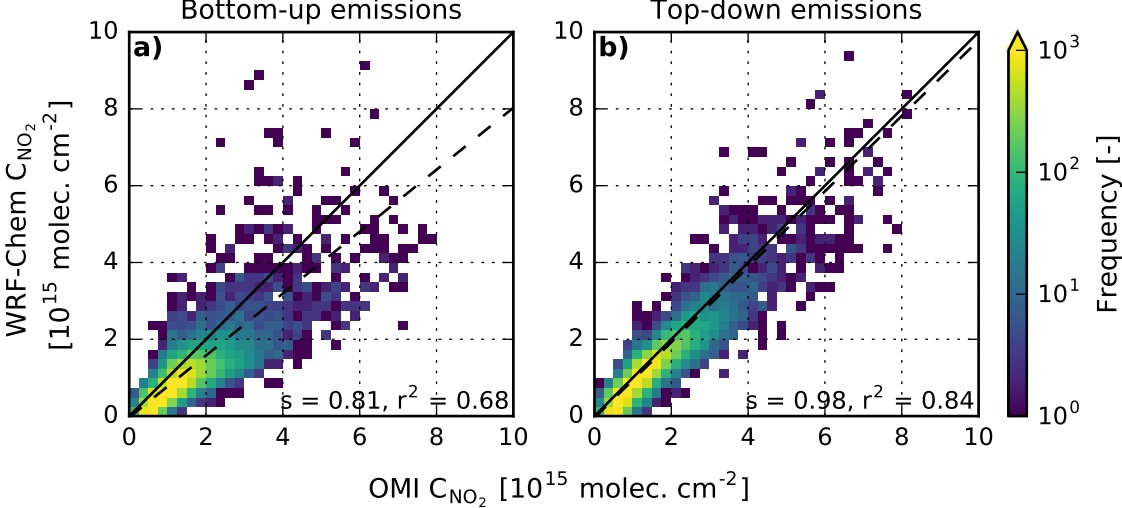

**Figure 6.** NO$_2$ vertical column density scatter plots of WRF-Chem against OMI, presented as a heat map with a bin size of $0.25 \times 10^{15}$ molec. cm$^{-2}$, for WRF-Chem with bottom-up emissions (a), and WRF-Chem with OMI-derived top-down surface NO$_x$ emissions (b).The OMI NO$_2$ VCDs in panels a) and b) are calculated with AMFs based on NO$_2$ vertical profiles of the WRF-Chem simulations against which they are compared, to ensure a consistent model-satellite comparison. The solid black lines represent the 1:1 line, and the dashed lines display the orthogonal distance regression fits.



**Table 2.** Comparison of WRF-Chem surface NO$_x$ emissions in July (in Tg N month$^{-1}$) with literature-reported values.

|  | Year | Region | Surface | Anthropogenic | Soils |
|---|---|---|---|---|---|
| This study, bottom-up | 2015 | Maps in this study | 0.32 | 0.30 | 0.015 |
| This study, top-down, after bias attribution (See Sect. 5.2) | 2015 | Maps in this study | 0.50 | 0.39-0.43 | 0.07-0.11 |
| Ganzeveld et al. (2010) | 2000 | -16-41°E, 34-64°N | - | - | 0.14 |
| Jaeglé et al. (2005) | 2000 | -15-45°E, 35-60°N | 0.59 | 0.35 | 0.25 |
| Miyazaki et al. (2017) | 2005-2014 | -10-30°E, 35-60°N | 0.33-0.38 | - | - |
| Dammers (2013) | 2005-2007 | -15-35°E, 35-70°N | - | - | 0.09 |
| Lathière et al. (2005) referenced in Dammers (2013) | 1983-1995 | -15-35°E, 35-70°N | - | - | 0.13 |





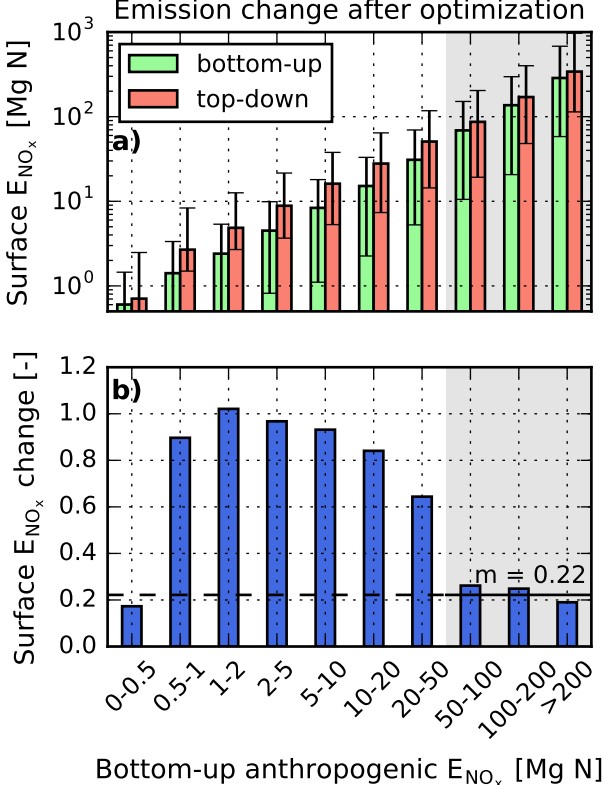

**Figure 7.** Difference between bottom-up and top-down surface $NO_x$ emissions, expressed as a) a bar plot (note the logarithmic scale) of median emissions binned by bottom-up anthropogenic $NO_x$ emissions (error bars indicate the inter-quartile range), and b) a bar plot of relative emission differences $\left( \frac{posterior - prior}{prior} \right)$ between the bars in panel a). In panel b) we define the relative anthropogenic emission difference to be the median of the relative change between top-down and bottom-up emissions in anthropogenic-dominated regions (shaded, with bottom-up emissions >50 Mg N month$^{-1}$ cell$^{-1}$.




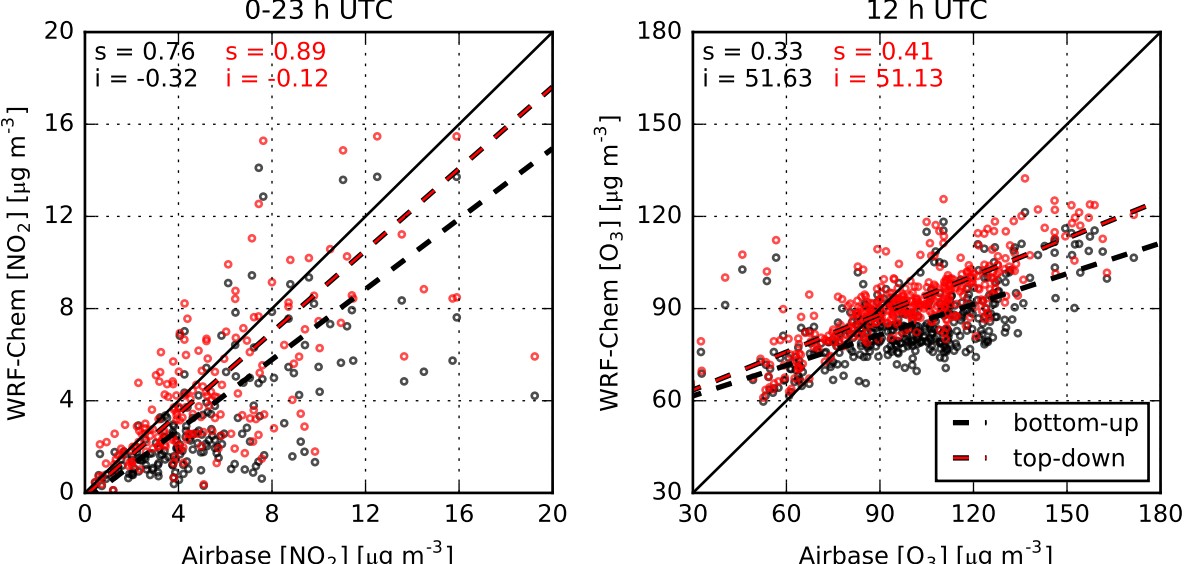

**Figure 8.** Scatter plots of monthly averaged simulated concentrations of a) $NO_2$ and b) $O_3$ against AirBase observations. Panel a) shows monthly averages for 0-23 h UTC, while panel b) is sampled at 12 h UTC. The black solid lines represent the 1:1 line.





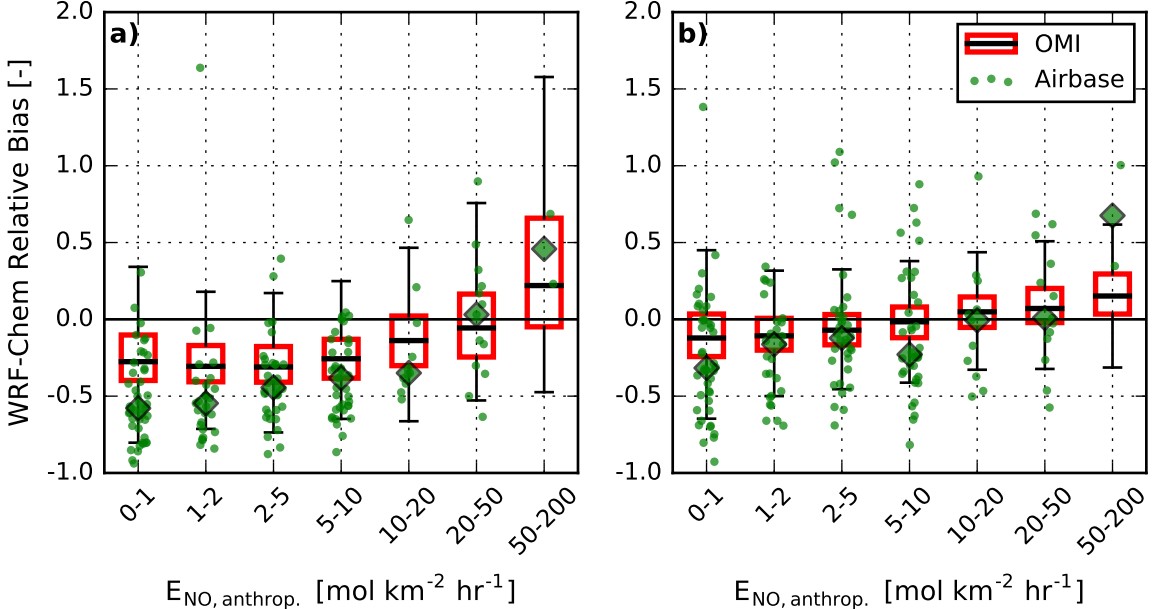

**Figure 9.** Relative bias $\left(\text{RB} = \frac{\text{model - observations}}{\text{observations}}\right)$ of WRF-Chem against land-based OMI NO$_2$ vertical column densities (box plots) and Air-Base in situ NO$_2$ measurements (green scatter), binned by bottom-up anthropogenic NO emission strength, for the bottom-up (a) and top-down WRF-Chem simulation (b). Green diamonds indicate the median WRF-Chem RB against AirBase observations for pixels within every emissions bin.




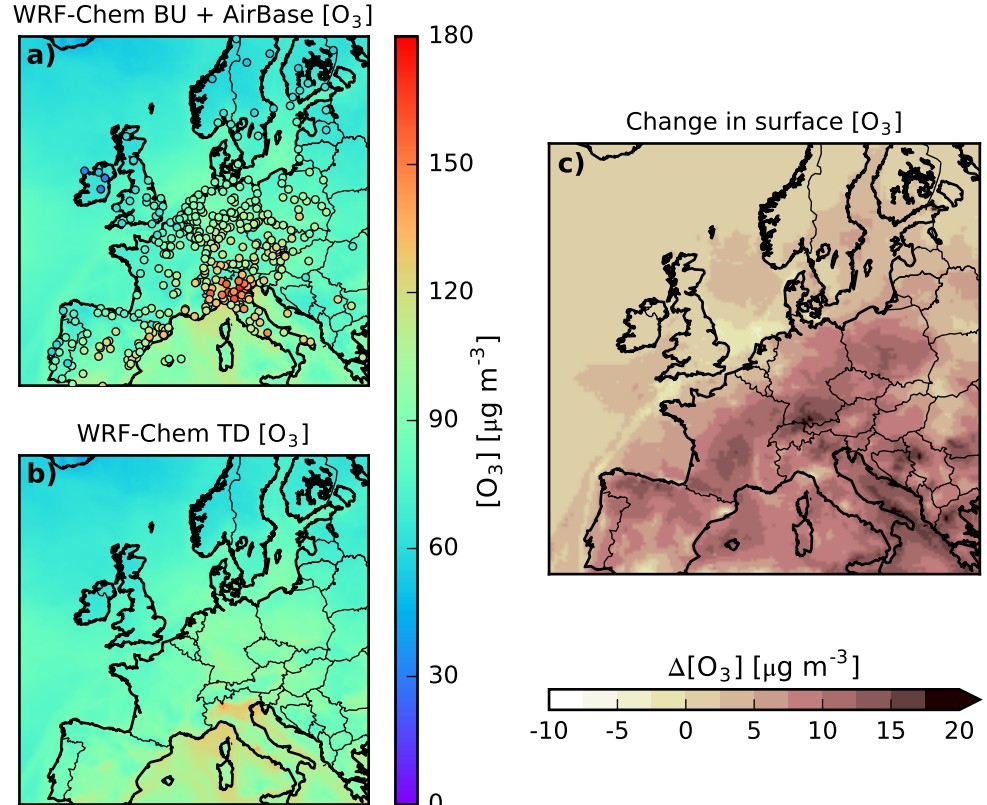

**Figure 10.** Monthly-averaged 12:00 h UTC surface $O_3$ concentration with bottom-up (BU, panel a) and top-down (TD, panel b) $NO_x$ emissions. Panel c shows the difference between the two monthly averages(TD - BU).





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
