# Peer review of "European NOx emissions in WRF-Chem derived from OMI"

_Atmospheric Chemistry and Physics, 2019_

## Short Comment (SC1) · 29 Apr 2019

This is an interesting study about European NOx emissions. I was puzzled by the fact that the ozone increase of 6 ug/m3 due to optimized emissions (which is reported to be largely due to increases in soil NOx emissions) is very similar to what I estimated as the impact of soil NOx emissions (albeit with even higher soil NOx emissions) of 4 (1.4-9.6) ppb a long time ago using a very simple all-European box model (Stohl, 1996). Is this just a coincidence, given the very simple set-up in Stohl (1996) and also differences in both soil and other NOx emissions between the two studies?

Reference: Stohl, A., E. Williams, G. Wotawa, and H. Kromp-Kolb (1996): A Euro-

pean inventory of soil nitric oxide emissions and the effect of these emissions on the photochemical formation of ozone in Europe. Atmos. Environ. 30, 3741-3755.

---

## Referee Comment (RC1) · Anonymous Referee #3 · 21 May 2019

I found the manuscript very well written and clear. All details of the methods seem to be explained in order to assure reproducibility and the results are logically and clearly illustrated. I think the manuscript is basically ready for publication, but I have only two comments/suggestions that the authors may evaluate for a minor revision:

- attribution to soil NOx emissions: the authors make a first-order estimate of the contribution of soil NOx emissions to increased total NOx emissions, after ingestion of satellite NO2 column data, using "anthropogenic" grid cells to estimate the contribution to NOx emissions from sources other than soils. This sounds to be reasonable, also considering the diffuse nature of the NOx emission change. A further relatively simple

test to confirm the hypothesis would be to run an additional simulation with increased bottom-up soil NOx emissions only by an x%, and see if the changes are consistent with the simulations using top-down emissions, both in terms of spatial distribution and magnitude.

- one interesting area is the Po Valley, which is the one showing the highest NO2 and O3 levels in the observations. The top-down correction of NOx emissions, however, does not seem to be effective enough in this area to fill the gap with observations. This point is sparsely discussed in the manuscript, but it would be useful to have some slightly further comment. For example, Figure 1 in the supplement shows that low values of beta (proportional to NO2 lifetime, from my understanding) are calculated upon main urban settlements (e.g. Milan), but the gamma factor (accounting for changes in the "shape" of the NO2 profile after update of emissions) is the lowest in Europe and pretty flat over the valley. Why is that and could this be a cause for the persistent underestimation of NOx emissions and O3 levels in the area? One rough idea is that the model possibly simulates a quite uniform PBL (thus a low gamma, from my understanding), even if this could be quite vertically inhomogeneous, due to recirculation of air in the valley (see e.g. Zhang and Rao (1999), J. Appl. Meteorol., 38, 1674–1691, doi:10.1175/1520-0450(1999)038<1674:TROVMI>2.0.CO;2; Ordonez et al. (2006) J. Geophys. Res., 111, D05310, doi:10.1029/2005JD006305; Curci et al. (2015) Atmos. Chem. Phys., 15, 2629-2649, https://doi.org/10.5194/acp-15-2629-2015). A further inspection in the vertical profiles over Po Valley, perhaps compared to other polluted regions such as Benelux would be instructive.
* * *

---

## Referee Comment (RC2) · Anonymous Referee #1 · 1 Jul 2019

The manuscript has presented the simulations of surface ozone concentrations over Europe in the regional air quality model (WRF-Chem). Its main focus is to analyse the changes in summertime surface ozone over Europe when replacing the bottom-up NOx emission inventories with top-down NOx emission estimates derived from the latest OMI NO2 product. The results show that OMI-constrained European NOx emissions are 56% higher than the bottom-up estimates, and that the increases can be largely attributed to large underestimates of agricultural soil emissions in the model. Model results with the top-down emissions significantly improve the comparison with surface in-situ NO2 measurements and moderately improve the comparison with surface ozone measurements as well.

[Figure]

Overall the manuscript is well organised and written, the methodology is sound. I recommend publish on ACP after the following comments been addressed.

**Specific Comments:**
(1) Page 1, Line 20 in the abstract:
What does "-48%" mean? Reduced by or to this value? Please clarify.

(2) Page 4, Line 17:
The study assumes 97% of NOx is emitted as NO and 3% as NO2. Can the model simulation of NO2 column be sensitive to this partitioning? Please discuss.

(3) Page 4, Line 31:
Here "+/-40%" should be "40%".

(4) Page 10, Line 17-20:
The sentence is confusing. Why the model underestimates of NO2 column would reflect emissions from power plants being too strong? Please clarify.

(5) Page 10, Line 25-28:
This statement did not explain why there was a larger model underestimate of surface NO2 concentration than that of NO2 column. Can you explain further? Would it reflect biases in model vertical transport or any measurement bias?

(6) Page 12, Line 20-22:
The sentence stated that model "underestimates the highest monthly averaged NO2 observations", but Figure 9 showed the opposite. Model results appeared to be slightly biases high for over high NOx emission regions.
And should hear 0.86 be 0.89?
(7) Page 14, Line 1:
Should 0.40 be 0.41 as seen from Table 1?

(8) Page 13, Section 6.2:
The improvement on surface ozone simulation with the top-down NOx emissions appears to be small. Can you also comment on some other metrics, such as time series of ozone levels at representative sites, or their diurnal cycles?

(9) Page 28, Figure 10: The right panel of Figure 10 is misleading by showing all values including negative values in red. Can you change the color table, e.g., use red for positive values, white for near-zero values, and blue for negative values?

---

## Author Comment (AC1) · 27 Aug 2019

**European NO$_x$ emissions in WRF-Chem derived from OMI: impacts on summertime surface ozone**

Submitted to: Atmospheric Chemistry and Physics

August 27, 2019

Auke J. Visser[1], K. Folkert Boersma[1,2], Laurens N. Ganzeveld[1], Maarten C. Krol[1,3]
auke.visser@wur.nl

[1]Wageningen University, Meteorology and Air Quality Section,
Wageningen, the Netherlands
[2]Royal Netherlands Meteorological Institute, R&D Satellite Observations,
de Bilt, the Netherlands
[3]Utrecht University, Institute for Marine and Atmospheric Research,
Utrecht, the Netherlands

**Contents**

**Abstract**

We would like to thank Anonymous Referees #1 and #3 for their useful and constructive feedback on our work. We additionally thank dr. Andreas Stohl for his comment. In this response letter, we carefully consider the issues that have been raised.

**1   Anonymous Referee #1**

> **Comment 1**
>
> The manuscript has presented the simulations of surface ozone concentrations over Europe in the regional air quality model (WRF-Chem). Its main focus is to analyse the changes in summertime surface ozone over Europe when replacing the bottom-up NOx emission inventories with top-down NOx emission estimates derived from the latest OMI NO2 product. The results show that OMI-constrained European NOx emissions are 56% higher than the bottom-up estimates, and that the increases can be largely attributed to large underestimates of agricultural soil emissions in the model. Model results with the top-down emissions significantly improve the comparison with surface in-situ NO2 measurements and moderately improve the comparison with surface ozone measurements as well.
>
> Overall the manuscript is well organised and written, the methodology is sound. I recommend publish on ACP after the following comments been addressed.

**Response**

We thank the Anonymous Referee #1 for her/his positive comments about our work and manuscript. We particularly thank the referee for the suggestion to include a discussion of time series and diurnal cycles, which we believe strengthens our message. Below, we address every comment carefully and explain the corresponding changes in the manuscript.

> **Comment 2**
>
> Page 1, Line 20 in the abstract: What does "-48%" mean? Reduced by or to this value? Please clarify.

**Response**

This should be: reduced by 48%. The sentence has been modified as follows (new text shown in blue):

> With respect to the initial simulation, MDA8 $O_3$ has an improved spatial distribution, expressed by an increase in $r^2$ from 0.40 to 0.53, and a decrease of the mean bias by 7.4 $\mu$g m$^{-3}$ (48%).

> **Comment 3**
>
> Page 4, Line 17: The study assumes 97% of NOx is emitted as NO and 3% as NO2. Can the model simulation of NO2 column be sensitive to this partitioning? Please discuss.

**Response**

Our $NO_x$ emission partitioning was motivated by the recommendation in the TNO-MACC-III anthropogenic emission dataset that 97% of $NO_x$ be emitted as NO, and the remaining 3% as $NO_2$. Indeed, the $NO_2/NO_x$ ratio in vehicle emissions, the largest $NO_x$ source in

Europe (Kuenen et al., 2014), is on the order of 10% to 20% (Carslaw, 2005). Surface concentrations can be sensitive to this emission ratio, as is seen at road-side air quality monitoring stations that are close to vehicle emission sources (Grange et al., 2017). Road-side observations sample air that has likely not reached photo-chemical equilibrium, so the $NO_x$ emission partitioning could be a potential source of error when comparing model output to observations from road-side stations. This motivates our choice to use only background air quality observations for our model comparison to in situ data.

However, the $NO_2$ column represents a vertically integrated amount of $NO_2$, composed of $NO_x$ emitted at the surface that has been transported horizontally and vertically, as well as $NO_x$ from adjacent model cells. We can assume that a photo-chemical equilibrium has been reached on the spatial scale of a $20\times20$ km$^2$ model pixel, especially in the model levels away from the lowermost level (but still in the boundary layer). Therefore, we do not think that the model-simulated $NO_2$ column is strongly sensitive to the $NO_x$ emission partitioning.

> **Comment 4**
>
> Page 4, Line 31: Here "+/-40%" should be "40%".

**Response**

Noted, the $\pm$ sign has been removed.

> **Comment 5**
>
> Page 10, Line 17-20: The sentence is confusing. Why the model underestimates of NO2 column would reflect emissions from power plants being too strong? Please clarify.

**Response**

We agree with the referee that this statement can be perceived as confusing. We therefore added extra context to further clarify this issue:

> For example, the simulated $NO_2$ column over northwestern Spain is underestimated by $2 \times 10^{15}$ molec. cm$^{-2}$ compared to OMI. The enhanced $NO_2$ columns in this region mainly reflect the contribution to atmospheric $NO_x$ by power plant emissions. Although emissions from power plants should have decreased in recent years in this region (Zhou et al., 2012), these emissions seem to be underestimated in WRF-Chem. However, since these results are only representative or July 2015, a more dedicated analysis is needed to further corroborate this hypothesis.

> **Comment 6**
>
> Page 10, Line 25-28: This statement did not explain why there was a larger model underestimate of surface NO2 concentration than that of NO2 column. Can you explain further? Would it reflect biases in model vertical transport or any measurement bias?

**Response**

Figure 9a in the main text indeed shows a lower model relative bias of surface $NO_2$ concentrations than of the tropospheric $NO_2$ column. We believe that missing surface $NO_x$ sources are responsible for the stronger underestimation at the surface. The surface model level, which is used for the comparison to surface stations, is more sensitive to missing $NO_x$ sources than the tropospheric $NO_2$ column.

We have investigated this while preparing the manuscript. We calculated the $\beta$-values (see Eqn. 3 in the main text) in two different ways: based on changes in the tropospheric $NO_2$ column, and based on changes in surface $NO_2$ concentrations (See Fig. 1). Column-based $\beta$-values are consistently higher, implying that a the emission increase needed to match column observations is larger than for in situ observations at the surface. This conclusion is supported by a study currently under discussion for ACP (Li and Wang, 2019) that also finds a stronger sensitivity for surface $NO_x$ emission changes at the surface compared to the tropospheric column.

[Figure]

Figure 1: Scatter plot of column $\beta$-values ($\beta_{\text{column}}$) versus surface $\beta$-values ($\beta_{\text{surface}}$), calculated by using a perturbation simulation with +20% surface emissions (see Sect. 3, Eq. 3). $\beta_{\text{surface}}$ is calculated in the same manner as $\beta_{\text{column}}$, using the surface $NO_2$ mixing ratio as the response variable.

We have modified the statement in the main text:

> There is a relatively larger model underestimation of surface $NO_2$ than of the $NO_2$ VCD in regions with comparatively low emissions. Given that the surface $NO_2$ mixing ratios are more sensitive to surface emissions than the $NO_2$ VCD (Li and Wang, 2019), this suggests that emissions are generally too low in WRF-Chem, but especially that emissions in rural background regions are underestimated.

**Comment 7**

Page 12, Line 20-22: The sentence stated that model "underestimates the highest monthly averaged NO2 observations", but Figure 9 showed the opposite. Model results appeared to be slightly biases high for over high NOx emission regions. And should hear 0.86 be 0.89?

**Response**

Firstly, thank you for noticing this typo, the slope should indeed be 0.89 instead of 0.86.

Figure 8 shows that, on average, WRF-Chem with top-down $NO_x$ emissions still slightly underestimates surface $NO_2$. However, Figure 9 shows that the model indeed overestimates the $NO_2$ column and surface $NO_2$ in regions with high $NO_x$ emissions (though there are only a limited number of AirBase background stations in these high-emission regions).

The results in Figure 8 are not in disagreement with those in Figure 9, because WRF-Chem overestimates the monthly-averaged surface $NO_2$ concentration at several stations. At those stations, WRF-Chem also shows a positive relative bias compared to AirBase in Figure 9. The improved slope between model and independent in situ observations indicates that our endeavour to derive satellite-based emissions has removed much of the systematic bias in simulated surface $NO_2$, but now leads to an overestimation in the simulated $NO_2$ concentrations at some stations. The scatter around the 1:1-line leads to low correlation values for $NO_2$ (also observed by e.g. Tuccella et al., 2012; Mar et al., 2016), and is likely caused by differences in spatial representativeness between a $20{\times}20$ km$^2$ model cell and in situ observations.

We have modified the sentence as follows, in order to ensure that the apparent contradiction between the results in Figures 8 and 9 is removed:

> The modified model set-up still slightly underestimates the  monthly-averaged $NO_2$ observations, as indicated by a slope of 0.89.

**Comment 8**

Page 14, Line 1: Should 0.40 be 0.41 as seen from Table 1?

**Response**

That is correct, this typo has been changed in the manuscript.

**Comment 9**

Page 13, Section 6.2: The improvement on surface ozone simulation with the top-down NOx emissions appears to be small. Can you also comment on some other metrics, such as time series of ozone levels at representative sites, or their diurnal cycles?

**Response**

We agree with Anonymous Referee #1 that the remaining model-observation mismatch is substantial. However, we actually believe a domain-average increase of the simulated monthly mean mid-day surface $O_3$ concentration of 6 μg m$^{-3}$ (in some regions reaching over 15 μg m$^{-3}$) after application of satellite-derived $NO_x$ is remarkable and agrees well with previously reported ozone sensitivities to $NO_x$ emissions (e.g Mallet and Sportisse, 2005; Li et al., 2019).

Nonetheless, we have taken up the suggestion by Anonymous Referee #1 to include an analysis of monthly diurnal cycles and time series of $O_3$ in six representative regions in Europe. This analysis supports our point that the afternoon ozone peak depends on $NO_x$ emissions. Therefore, the main text now contains a discussion of $O_3$ diurnal cycles, whereas the complete time series are included in the supplement.

The six representative regions span several degrees in latitude/longitude and contain 18-59 stations for a time series comparison (Figs. 2 and 3, and Table 1). Two have high $NO_x$ emissions (BeNeLux + Ruhr, Po Valley), with contrasting ozone production efficiency due to temperature and radiation differences. Two sites are situated in low-$NO_x$ rural background areas (Central France and Central Spain), while two other regions show a considerable $NO_x$ emission increase that apparently results in a strong response in $O_3$ (Southern Germany, Poland).

The following section has been added to the main text:

> We additionally analyzed changes in the temporal evolution of ozone concentrations resulting from $NO_x$ emission changes (Fig. 10). Daytime median $O_3$ concentrations are better captured in the Po Valley, Central Spain and Poland. The $NO_x$ emission changes lead to a model overestimation of surface $O_3$ concentrations for Central France and South Germany, while concentrations change only slightly in the BeNeLux and Ruhr areas. In those regions, the mean bias error increases, while the hourly correlation coefficient and RMSE values improve for all regions (Supplementary Table 4). In all areas, changes in $NO_x$ emissions lead to increased ozone concentrations particularly during daytime. Enhancements in simulated night-time concentrations are only observed in Central Spain. In other areas, night-time $O_3$ concentrations are overestimated in both simulations. Peak daytime $O_3$ concentrations are better captured in all areas, as evidenced by the increase of the 75$^{th}$ percentile of simulated $O_3$ concentrations with top-down emissions. However, peak $O_3$ concentrations remain underestimated in the Po Valley, Central Spain and South Germany. Additionally, nighttime $O_3$ concentration overestimations remain, likely due to issues related to model resolution and vertical mixing. Overall, the $NO_x$ emission changes most effectively increase $O_3$ concentrations during periods with elevated ozone (Fig. S3), which coincide with high solar radiation and temperatures and thus have a strongly $NO_x$-dependent $O_3$ formation.

[Figure]

Figure 2: July 2015 monthly median diurnal ozone concentrations for six representative regions in Europe, as simulated by WRF-Chem with a priori $NO_x$ emissions (green line) and a posteriori $NO_x$ emissions (red line), and as observed at AirBase stations in these regions. Shaded areas and whiskers indicate the inter-quartile range. Results represent the median over all model-observation comparisons per region. The sample size for the comparison is displayed on the top right of each subplot.

Table 1: Model performance statistics for surface ozone concentration time series of the WRF-Chem simulation with bottom-up and top-down emissions for six European regions.

|  | Po Valley | BeNeLux + Ruhr | Central France | Central Spain | South Germany | Poland |
|---|---|---|---|---|---|---|
| n (stations) | 59 | 32 | 29 | 24 | 39 | 18 |
| | | | Bottom-up | | | |
| MBE | -20.14 | 16.82 | 3.35 | -22.40 | -11.15 | 1.74 |
| RMSE | 68.07 | 71.48 | 59.92 | 45.39 | 68.68 | 43.64 |
| r | 0.80 | 0.78 | 0.76 | 0.81 | 0.74 | 0.77 |
| | | | Top-down | | | |
| MBE | -1.58 | 25.94 | 17.29 | -1.33 | 5.02 | 16.10 |
| RMSE | 55.08 | 68.57 | 56.48 | 36.44 | 58.13 | 41.81 |
| r | 0.85 | 0.81 | 0.79 | 0.83 | 0.81 | 0.80 |

[Figure]

Figure 3: July 2015 time series of the median O₃ concentrations as observed at AirBase stations (black dots), and as simulated by WRF-chem with bottom-up (red) and top-down emissions (green). Medians are calculated by including all stations (resp. co-sampled simulations) in the latitude/longitude range specified in the subplot titles. Shaded areas show the inter-quartile range.

> ### Comment 10
>
> Page 28, Figure 10: The right panel of Figure 10 is misleading by showing all values including negative values in red. Can you change the color table, e.g., use red for positive values, white for near-zero values, and blue for negative values?

**Response** We agree that a diverging colormap is more appropriate here. The colormap has been updated in the figure.

[Figure]

Figure 4: Monthly-averaged 12:00 h UTC surface $O_3$ concentration with bottom-up (BU, panel a) and top-down (TD, panel b) $NO_x$ emissions. Panel c shows the difference between the two monthly averages (TD - BU).

**2   Anonymous Referee #3**

> **Comment 1**
>
> I found the manuscript very well written and clear. All details of the methods seem to be explained in order to assure reproducibility and the results are logically and clearly illustrated. I think the manuscript is basically ready for publication, but I have only two comments/suggestions that the authors may evaluate for a minor revision:

**Response**

We thank Anonymous Referee #3 for her/his positive evaluation of our study. The suggestions for further analysis are interesting, and we address them below.

> **Comment 2**
>
> Attribution to soil NOx emissions: the authors make a first-order estimate of the contribution of soil NOx emissions to increased total NOx emissions, after ingestion of satellite NO2 column data, using "anthropogenic" grid cells to estimate the contribution to NOx emissions from sources other than soils. This sounds to be reasonable, also considering the diffuse nature of the NOx emission change. A further relatively simple test to confirm the hypothesis would be to run an additional simulation with increased bottom-up soil NOx emissions only by an x%, and see if the changes are consistent with the simulations using top-down emissions, both in terms of spatial distribution and magnitude.

**Response**

Indeed, a strong and uniform increase in soil $NO_x$ emissions would lead to increases in simulated peak ozone concentrations, and would therefore reduce model bias in rural areas: a recent study found an increase in the monthly-averaged daily maximum ozone mixing ratio of 6 ppb after increasing a priori emissions from soils by 500% (Li et al., 2019). A sensitivity test that we performed while preparing the manuscript, in which soil $NO_x$ emissions were uniformly scaled up by 86%, points in the same direction. We are however hesitant to include a sensitivity study with a uniform scaling factor for soil $NO_x$ emissions in the paper, since this goes against our point that a strong contribution by fertilizer application (Ganzeveld et al., 2010) is likely missing in the a priori soil $NO_x$ emission budget, leading to a wrong spatial distribution of soil $NO_x$ emissions. The contribution of fertilizer-induced $NO_x$ emissions in Europe varies strongly per country (e.g. Butterbach-Bahl et al., 2009). The sensitivity of (peak) ozone concentrations to soil $NO_x$ emissions is further reflected by studies introducing improvements in the process-based representation of soil $NO_x$ emissions in the CMAQ CTM, which found strong increases of MDA8 $O_3$ and a reduced mean ozone concentration bias over agricultural areas (Rasool et al., 2016, 2019).

The study by Li et al. (2019), which exactly describes the experiment that Anonymous Referee #3 requests, albeit for a different study area, was not yet published at the time of submission. We have therefore added the following content to our discussion:

Several studies previously investigated the relation between soil $NO_x$ emissions and $O_3$ formation. For example, one study estimated that European soil $NO_x$ emissions contribute 4 ppb to the daily maximum concentration (Stohl et al., 1996). A sensitivity study by Li et al. (2019) indicates that a strong up-scaling of soil $NO_x$ emissions by a factor 5 indeed leads to a better representation of the peak ozone concentration. It has further been shown that an improved process-based representation of soil $NO_x$ emissions leads to MDA8 $O_3$ changes by up to 6 ppb (Rasool et al., 2016), and a reduced mean bias for ozone concentrations, particularly in agricultural areas (Rasool et al., 2019). Together, these findings provide support for the hypothesis that underestimated soil $NO_x$ emissions, in particular those from agricultural areas, contribute to underestimated peak ozone concentrations.
* * *
**Comment 3**

One interesting area is the Po Valley, which is the one showing the highest NO2 and O3 levels in the observations. The top-down correction of NOx emissions, however, does not seem to be effective enough in this area to fill the gap with observations. This point is sparsely discussed in the manuscript, but it would be useful to have some slightly further comment. For example, Figure 1 in the supplement shows that low values of beta (proportional to NO2 lifetime, from my understanding) are calculated upon main urban settlements (e.g. Milan), but the gamma factor (accounting for changes in the "shape" of the NO2 profile after update of emissions) is the lowest in Europe and pretty flat over the valley. Why is that and could this be a cause for the persistent underestimation of NOx emissions and O3 levels in the area? One rough idea is that the model possibly simulates a quite uniform PBL (thus a low gamma, from my understanding), even if this could be quite vertically inhomogeneous, due to recirculation of air in the valley (see e.g. Zhang and Rao (1999), J. Appl. Meteorol., 38, 1674-1691, doi:10.1175/1520-0450(1999)038<1674:TROVMI>2.0.CO;2; Ordonez et al. (2006) J. Geophys. Res., 111, D05310, doi:10.1029/2005JD006305; Curci et al. (2015) Atmos. Chem. Phys., 15, 2629-2649, https://doi.org/10.5194/acp-15-2629-2015). A further inspection in the vertical profiles over Po Valley, perhaps compared to other polluted regions such as Benelux would be instructive.
* * *
**Response**

Firstly, we do not agree with Anonymous Referee #3 that $NO_x$ emissions and $O_3$ concentrations are underestimated in the Po Valley in the simulation with top-down emissions. Our approach to re-calculate $NO_x$ emissions based on OMI data leads to an almost 1:1 agreement with OMI (the dependent variable, indicating that biases are effectively removed using our approach) at an $r^2$ of 0.84, and a better agreement with surface $NO_2$ observations. Additionally, in Fig. 2 (Fig. 10 in the revised manuscript) we show that median ozone levels in the Po Valley in the simulation with top-down $NO_x$ emissions agree well with observations, although peak ozone concentrations remain underestimated.

However, we thank Anonymous Referee #3 for raising this interesting point regarding the effect of $\gamma$ on $NO_x$ emissions. It is correct that $\gamma$ expresses the sensitivity of the AMF to a change in the $NO_2$ profile shape resulting from $NO_x$ emission changes (Vinken et al., 2014).

In simple terms, the air mass factor (AMF) can be understood as the convolution of the averaging kernel, expressing the decreasing vertical sensitivity of the instrument (Eskes and Boersma, 2003) towards the surface (in cloud-free conditions), and the vertical $NO_2$ profile. The relative change in the profile shape is lower for high-emission areas compared to low-emission areas, since high-emission regions already have a strongly peaked $NO_2$ profile at the surface. A +20% perturbation in $NO_x$ emissions, as used to calculate $\gamma$, will lead to a relatively lower increase of $NO_2$ levels in high-$NO_x$ regions compared to low-$NO_x$ areas. The resulting change in the AMF is low in high-emission areas such as the Po Valley, the BeNeLux and the Ruhr area. This explains the low $\gamma$ values in polluted regions.

Lastly, suppose that vertical re-circulation of $NO_x$- and $O_3$-rich air (as can be seen in the references brought forward by Anonymous Referee #3) is indeed underestimated in WRF-Chem. More efficient vertical mixing conditions would then lead to a less strongly peaked monthly-average $NO_2$ profile at the surface following a +20% emission perturbation. The OMI $NO_2$ column will display a relatively lower increase following AMF updates compared to a case with lower vertical mixing. Therefore, following Eqn. 4 in the main text, this will lead to *lower* $\gamma$-values and thus to a less strong emission update.

**3   Dr. Andreas Stohl**

> **Comment 1**
>
> This is an interesting study about European NOx emissions. I was puzzled by the fact that the ozone increase of 6 ug/m3 due to optimized emissions (which is reported to be largely due to increases in soil NOx emissions) is very similar to what I estimated as the impact of soil NOx emissions (albeit with even higher soil NOx emissions) of 4 (1.4-9.6) ppb a long time ago using a very simple all-European box model (Stohl, 1996). Is this just a coincidence, given the very simple set-up in Stohl (1996) and also differences in both soil and other NOx emissions between the two studies?

**Response**

We would like to thank Dr. Stohl for his interest, his positive words about our study and his question.

Given the large differences in approach between the current study and Stohl et al. (1996), concerning model complexity and input emission datasets, the similarity in ozone sensitivity to (soil) $NO_x$ emissions could be coincidental. Nonetheless, the study by Stohl et al. (1996) provides support for our hypothesis that mis-representations in soil $NO_x$ emissions, particularly from land management practices, can contribute to biased ozone simulations. The one-dimensional model that is used in the study is likely to generally represent $NO_x$-limited conditions during daytime, meaning that an addition of $NO_x$ from soils leads to efficient $O_3$ production.

The annual emission totals reported by Stohl et al. (1996) are indeed higher than ours, but are difficult to compare to our results without accurate knowledge on the seasonal cycle in emissions. However, soil emissions are 21.4% of combustion-related 'pyrogenic' emissions in summer (JJA) 1994 in Stohl et al. (1996), which amounts to 17.6% of total surface emissions (soils + pyrogenic). We derive soil emissions to be 14-22% of total European $NO_x$ emissions in July 2015, suggesting that our top-down estimates are reasonable.

Since the study brought to our attention by Dr. Stohl provides support for our satellite-based soil $NO_x$ emission estimate, we added a reference to Stohl et al. (1996) to Table 2 in the manuscript. We additionally added a reference to Stohl et al. (1996) in a new section in the discussion (see response to Anonymous Referee #3, comment 2), which places our findings regarding the ozone increases and our attribution to soil $NO_x$ in a literature context:

> Several studies previously investigated the relation between soil $NO_x$ emissions and $O_3$ formation. For example, one study estimated that European soil $NO_x$ emissions contribute 4 ppb to the daily maximum concentration (Stohl et al., 1996). A sensitivity study by Li et al. (2019) indicates that a strong up-scaling of soil $NO_x$ emissions by a factor 5 indeed leads to a better representation of the peak ozone concentration. It has further been shown that an improved process-based representation of soil $NO_x$ emissions leads to MDA8 $O_3$ changes by up to 6 ppb (Rasool et al., 2016), and a reduced mean bias for ozone concentrations, particularly in agricultural areas (Rasool et al., 2019). Together, these findings

provide support for the hypothesis that underestimated soil $NO_x$ emissions contribute to underestimated peak ozone concentrations.

**References**

Butterbach-Bahl, K., Kahl, M., Mykhayliv, L., Werner, C., Kiese, R., and Li, C. (2009). A European-wide inventory of soil NO emissions using the biogeochemical models DNDC/Forest-DNDC. Atmospheric Environment, 43(7):1392–1402.

Carslaw, D. C. (2005). Evidence of an increasing NO2/NOX emissions ratio from road traffic emissions. Atmospheric Environment, 39(26):4793–4802.

Eskes, H. J. and Boersma, K. F. (2003). Averaging kernels for DOAS total-column satellite retrievals. Atmospheric Chemistry and Physics, 3(5):1285–1291.

Ganzeveld, L., Bouwman, L., Stehfest, E., van Vuuren, D., Eickhout, B., and Lelieveld, J. (2010). Impacts of future land cover changes on atmospheric chemistry-climate interactions. Journal of Geophysical Research, 115.

Grange, S. K., Lewis, A. C., Moller, S. J., and Carslaw, D. C. (2017). Lower vehicular primary emissions of NO 2 in Europe than assumed in policy projections. Nature Geoscience, 10(2).

Kuenen, J. J. P., Visschedijk, A. J. H., Jozwicka, M., and Denier van der Gon, H. a. C. (2014). TNO-MACC_II emission inventory: a multi-year (2003-2009) consistent high-resolution European emission inventory for air quality modelling. Atmos. Chem. Phys., 14(2013):10963–10976.

Li, J. and Wang, Y. (2019). Inferring the anthropogenic NOx emission trend over the United States during 2003-2017 from satellite observations: Was there a flattening of the emission tend after the Great Recession? Atmospheric Chemistry and Physics Discussions, (July):1–35.

Li, J., Wang, Y., and Qu, H. (2019). Dependence of Summertime Surface Ozone on NOx and VOC Emissions Over the United States: Peak Time and Value. Geophysical Research Letters, 46(6):3540–3550.

Mallet, V. and Sportisse, B. (2005). A comprehensive study of ozone sensitivity with respect to emissions over Europe with a chemistry-transport model. Journal of Geophysical Research Atmospheres, 110(22):1–15.

Mar, K. A., Ojha, N., Pozzer, A., and Butler, T. M. (2016). Ozone air quality simulations with WRF-Chem ( v3 . 5 . 1 ) over Europe : model evaluation and chemical mechanism comparison. Geoscientific Model Development, 9:3699–3728.

Rasool, Q. Z., Bash, J. O., and Cohan, D. S. (2019). Mechanistic representation of soil nitrogen emissions in the Community Multiscale Air Quality (CMAQ) model v 5.1. Geoscientific Model Development, 12(2):849–878.

Rasool, Q. Z., Zhang, R., Lash, B., Cohan, D. S., Cooter, E. J., Bash, J. O., and Lamsal, L. N. (2016). Enhanced representation of soil NO emissions in the Community Multiscale Air Quality (CMAQ) model version 5.0.2. Geoscientific Model Development, 9(9):3177–3197.

Stohl, A., Williams, E., Wotawa, G., and Kromp-Kolb, H. (1996). A European inventory of soil nitric oxide emissions and the effect of these emissions on the photochemical formation of ozone. Atmospheric Environment, 30(22):3741–3755.

Tuccella, P., Curci, G., Visconti, G., Bessagnet, B., Menut, L., and Park, R. J. (2012). Modeling of gas and aerosol with WRF/Chem over Europe: Evaluation and sensitivity study. Journal of Geophysical Research Atmospheres, 117(3):1–15.

Vinken, G. C. M., Boersma, K. F., van Donkelaar, A., and Zhang, L. (2014). Constraints on ship NOx emissions in Europe using GEOS-Chem and OMI satellite NO2 observations. Atmospheric Chemistry and Physics, 14:1353–1369.

Zhou, Y., Brunner, D., Hueglin, C., Henne, S., and Staehelin, J. (2012). Changes in OMI tropospheric NO 2 columns over Europe from 2004 to 2009 and the in fl uence of meteorological variability. Atmospheric Environment, 46(2):482–495.